# Cell-attribute aware community detection improves differential abundance testing from single-cell RNA-Seq data

Alok K. Maity[1] & Andrew E. Teschendorff [1,2] ✉

Variations of cell-type proportions within tissues could be informative of biological aging and disease risk. Single-cell RNA-sequencing offers the opportunity to detect such differential abundance patterns, yet this task can be statistically challenging due to the noise in single-cell data, inter-sample variability and because such patterns are often of small effect size. Here we present a differential abundance testing paradigm called ELVAR that uses cell attribute aware clustering when inferring differentially enriched communities within the single-cell manifold. Using simulated and real single-cell and single-nucleus RNA-Seq datasets, we benchmark ELVAR against an analogous algorithm that uses Louvain for clustering, as well as local neighborhood-based methods, demonstrating that ELVAR improves the sensitivity to detect cell-type composition shifts in relation to aging, precancerous states and Covid-19 phenotypes. In effect, leveraging cell attribute information when inferring cell communities can denoise single-cell data, avoid the need for batch correction and help retrieve more robust cell states for subsequent differential abundance testing. ELVAR is available as an open-source R-package.

Detecting shifts of cell-type proportions in relation to aging, exposures or disease risk factors is an important task to improve our understanding of disease predisposition and disease onset[1]. Single-cell technologies, and single-cell RNA sequencing (scRNA-Seq)[2] in particular, offer the opportunity to detect such differential abundance (DA) patterns, but this task can be statistically challenging[3]. One key challenge is that, almost inevitably, any assessment of DA across biological conditions entails a comparison of cell-type numbers from assays performed in different subjects. Thus, biological inter-subject variability, as well as technical batch effects, can potentially confound naïve DA-analyses[4]. Technical batch effects can be addressed by performing scRNA-Seq assays in a number of different subjects representing the same biological condition, called sample replicates. Using such biological replicates also helps gauge the biological inter-sample variability, which can be substantial. For instance, it is now well recognized that different individuals may age at different rates[5], and that such

variations in biological age may be associated with underlying shifts in T-cell proportions[6]. Thus, taking inter-sample variability into account is critically important when testing for DA. Another major challenge is that DA-patterns of interest are often sought across biological conditions (e.g. aging or exposure to other disease risk factors) that only induce relatively small shifts in the proportions of very similar cell subtypes. While scRNA-Seq technology and standard clustering approaches allow relatively easy discrimination of major cell-types (e.g. fibroblasts, epithelial cells, T-cells), the discrimination of cell subtypes such as naïve vs memory T-cells, or the discrimination of cells according to a biological condition such as age or disease risk, is more challenging due to the small effect sizes involved and the noisy nature of single-cell data. Thus, it is critically important to devise methods that can robustly identify relevant cell-states in the background of such noisy data, to ensure that the subsequent quantification of DA-patterns is reliable.

[1]CAS Key Laboratory of Computational Biology, Shanghai Institute of Nutrition and Health, University of Chinese Academy of Sciences, Chinese Academy of Sciences, 320 Yue Yang Road, Shanghai 200031, China. [2]UCL Cancer Institute, Paul O'Gorman Building, University College London, 72 Huntley Street, London WC1E 6BT, United Kingdom. ✉e-mail: andrew@sinh.ac.cn

In response to these challenges, various statistical DA-testing algorithms have been proposed[7–12], with some of the more recent methods (e.g. refs. [10–12]) taking sample replication into account. DA-testing algorithms also differ in terms of whether they use the full single-cell state manifold[13,14] when inferring DA or not. At one extreme we have DA-testing methods that only rely on the discrete cell clusters and cell-type annotation derived from the scRNA-Seq data[1,3,11,12,15], whilst at the other extreme we have methods that make full use of the manifold structure[7,10]. These latter studies have advocated the need to estimate DA from fuzzier representations of cell clusters, known as cellular neighborhoods, in recognition of the fact that cells generally only cluster by broad cell-types and not by underlying biological conditions or cell subtypes[7,8,10]. Indeed, although clustering-based approaches have been successful in identifying cell subtypes or rare cell-types[3], the defining characteristic of these subtypes is their low proportion and moderate effect sizes, whilst DA-testing is often needed in the context of more abundant cell subtypes defined by much smaller effect sizes. We reasoned that if cell clustering algorithms could be generalized to take cell attribute information into account, for instance, a cell's biological condition such as age or disease stage, that this would improve the signal-to-noise ratio, and, in so doing, allow retrieval of more biologically relevant clusters and robust cell-states. In support of our hypothesis, we note that clustering graph nodes by taking node-attributes into account has been a fruitful approach in network science generally[16]. Hence, this node-attribute-aware clustering paradigm ought to be beneficial for tackling the uncertainties and noise associated with single-cell omic data. Moreover, cell attributes themselves often encode information that is part of the intrinsic process that generates cell communities, so not using such information in noisy data may lead to incomplete or imprecise clusters. A final reason to consider cell-attribute aware clustering is that it can help discern cell communities that are shared across sample replicates, and which are therefore more likely to be biologically relevant.

To test our hypothesis, we here adapt a node-attribute aware community detection algorithm called EVA[17], which is a generalization of the very popular Louvain clustering method[18]. Of note, despite Louvain's great popularity in the single-cell analysis field[3], the Louvain algorithm only takes the topology of the cell-cell similarity graph into account when inferring cell communities. EVA can be viewed as a direct extension of the Louvain algorithm allowing multiple cell attributes to be incorporated when inferring communities. This novel concept allows clustering of cells, not only by similarity in the high-dimensional state-space, but also by how similar their attributes (e.g. age, disease stage) are. Here we develop a novel R-implementation of EVA and incorporate it into a complete algorithmic pipeline for DA-testing called ELVAR. We subsequently validate ELVAR very extensively on both simulated as well as real datasets, demonstrating improved sensitivity over competing methods.

## Results

### Rationale of ELVAR algorithm

Detecting shifts in cell-subtype proportions across different biological conditions from scRNA-Seq data can be challenging when these factors drive only a small fraction of the overall data variance. Indeed, with scRNA-Seq data, cells generally cluster by the main cell-types present in a tissue, for instance, epithelial, immune and fibroblast types. However, more refined clusterings that clearly discriminate cells according to subtypes (e.g. different CD4+ T-cell subtypes) or biological conditions that only cause a relatively small change in the transcriptome of cells (e.g. age of a cell) are less forthcoming. In effect, this challenge arises whenever relevant components of biological variation carry similar or less variance compared to technical factors, thus preventing the segregation of cells by biological condition or subtype. We hypothesized that DA-testing would benefit from a procedure that can more reliably identify relevant cellular states and that this could be achieved if clustering analyses were to include cell-attributes into account when inferring cellular communities. Specifically, following standard dimensional reduction and inference of a cell-cell nearest neighbor graph (Fig. 1a)[19], we posited that more relevant cell communities could be identified if the inference of clusters in this graph were to use cell-attribute information, since this would favor clustering solutions where cells within communities are predominantly of one biological condition or subtype. To test our hypothesis, we adapt an extension of the popular Louvain algorithm, called EVA[17], which unlike Louvain, aims to maximize an objective function that includes a purity index besides modularity. This purity index measures how homogeneous inferred communities are in relation to some particular cell-attribute (e.g. age or disease stage). Henceforth, we shall refer to the cell-attribute used in the purity calculation as the "clustering attribute". The EVA algorithm also includes a purity parameter "a" that controls the relative importance of purity over modularity when inferring communities (Fig. 1b). For a given cell-attribute value, communities enriched for cells taking on that value can subsequently be identified (Fig. 1c). Of note, whilst these communities may contain cells from multiple sample replicates, we do not impose this, safeguarding flexibility and power. If cells have an additional attribute, called the attribute of interest (e.g. differentiation-state or cell-subtype), negative binomial regressions (NBRs) can then be used to assess if the proportions of this attribute of interest changes across the clustering attribute representing distinct biological conditions (i.e. age or disease stage)[10] (Fig. 1d). We implement the whole DA-testing pipeline (including the cell-attribute-aware clustering and NBR steps) in the programming language R, calling the resulting algorithm ELVAR (Extended LouVain Algorithm for DA-testing in R)[20].

### Validation of EVA and ELVAR on simulated data

In order to test our R-implementation of EVA, we devised a simulation model based on scRNA-Seq data from the Tabula Muris Senis (TMS)[21], consisting of 200 scRNA-Seq profiles representing classical monocytes from one particular mouse, with 100 cells defining a perturbed state (P) and the remaining 100 representing an unperturbed normal (N) condition (Methods). We simulated differences in gene expression between the two conditions to be subtle, only involving 0.2% of all genes. Dimensional reduction and visualization with t-stochastic neighborhood embedding (t-SNE) did not reveal any clustering structure except for the distinctively non-random distribution of perturbation-state within the main cluster (Supplementary Fig. S1a). Louvain clustering over the cell-cell nearest neighbor graph in the higher dimensional state manifold revealed a more complex clustering structure with nine clusters that globally correlated with perturbation state (Supplementary Fig. S1a). Applying EVA for a range of different purity parameter values (Supplementary Fig. S1b) revealed stronger correlations with perturbation state, as evaluated using the adjusted Rand Index (ARI) (Supplementary Fig. S1c) or with Chi-Square statistic $P$-values (Supplementary Fig. S1a), albeit only for larger purity parameter values. Restricting to EVA solutions with the same number of inferred communities as Louvain ($n = 9$), also displayed improved ARI and more significant chi-square statistic $P$-values compared to Louvain (Supplementary Fig. S1d). We verified that the improvement of EVA over Louvain was independent of the resolution parameter, which, in addition to the purity parameter, also controls the number and size of inferred communities (Methods, Supplementary Fig. S2).

To validate ELVAR, we generalized the above simulation model to include cells from different age groups and multiple mouse replicates, and by increasing the frequency of the perturbation state between young and old groups in order to simulate age-related differential abundance (Fig. 2a). Using a well-defined set of criteria that balances purity, modularity and cluster number (Methods), we identified a purity parameter value $a = 0.8$ as the optimal choice for EVA in this

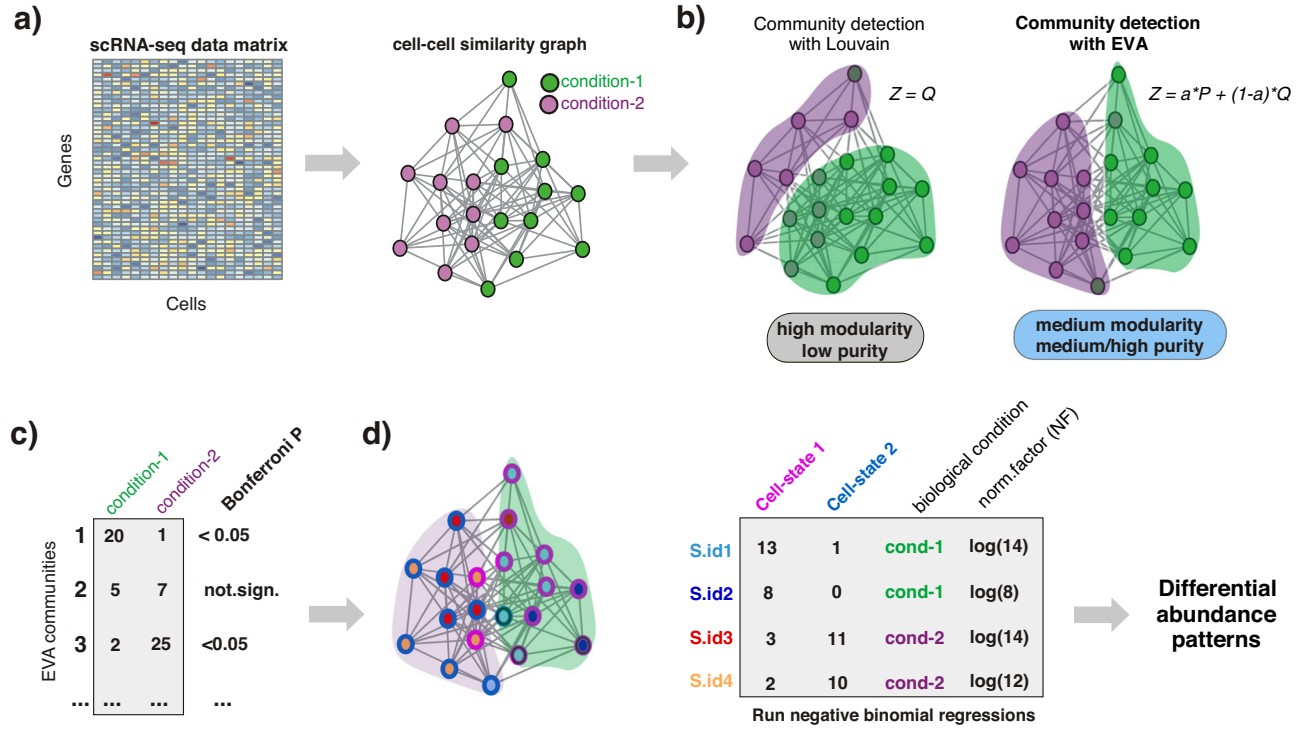

**Fig. 1 | Flowchart of ELVAR algorithm. a** Given a scRNA-Seq data matrix with cells derived from various conditions (e.g. age-groups), one first derives a cell-cell similarity graph using standard pipelines like Seurat. Cells may also differ in terms of an attribute of interest (e.g. cell state or cell subtype) and the sample replicate it is derived from. **b** To infer communities from this cell-cell graph, we use an extended Louvain algorithm (EVA) which, unlike the standard Louvain algorithm, takes cell attribute information into account when deriving the communities. In this case, the cell-attribute used in the clustering (the clustering attribute) could be the biological condition it is derived from, in which case the inferred communities will be more enriched for cells of the same condition, as shown. Compared to the standard Louvain algorithm, which aims to maximize the overall modularity $Q$ of the dataset (Fig. 2b). EVA inferred enriched age-related communities from the nearest neighbor cell-cell graph (Fig. 2c), and using negative binomial regressions to account for inter-mouse replicate variation, we correctly inferred the expected increase of the perturbation state fraction with age (Fig. 2d, e). Importantly, statistical significance attained by ELVAR was stronger compared to analogous algorithms that use either the sequential (deterministic) or nonsequential (stochastic) Louvain algorithm instead of EVA (Fig. 2e, Methods). Of note, whilst this improvement comes at the expense of a higher computational complexity, EVA/ELVAR runtimes are feasible for reasonably sized cell-cell networks (Supplementary Fig. S3). For instance, for a network with approximately $n = 30,000$ cells, EVA/ELVAR runtime (1 run) on a typical professional workstation takes around 15–20 min.

the communities, EVA aims to maximize a weighted sum of $Q$ and the overall purity $P$ (a measure of how pure the communities are in relation to the conditions). The $a$ parameter controls the relative importance of $Q$ and $P$ when maximizing the objective function $Z$. **c** EVA communities that are significantly enriched for cells from a particular condition are selected for further downstream analysis, thus removing noisy cellular neighborhoods. **d** For a given condition, cells from all communities enriched for that condition are merged and the distribution of underlying cell-states from each sample replicate are computed. Finally, negative binomial regressions are used to infer if given cell-state fractions (the attribute of interest) vary significantly with condition, whilst taking sampling variability into account.

## ELVAR improves sensitivity to detect DA-shifts in real scRNA-Seq datasets

Having demonstrated ELVAR's improved sensitivity for DA-testing on simulated scRNA-Seq data, we next aimed to validate ELVAR on real data, and to test if similar improvements over the Louvain-benchmark are also seen in real data. We first considered the case of aging in Cd4+ T-cells. As described by several studies, the naïve subset of Cd4+ T-cells in blood decreases with age, contributing to the well-known phenomenon of immuno-senescence[6,22]. Hence, ELVAR should be able to predict an analogous age-related shift from naïve Cd4+ T-cells to the more mature subtype in tissues with significant amount of immune-cell infiltration, such as lung[23]. To this end, we considered the lung-tissue 10X scRNA-Seq dataset from the Tabula Muris Senis[21], due to ample profiling of Cd4+ T-cells in this tissue across at least 5 age groups, ranging from one-month (1 m) to 30 month-old mice (30 m). After QC, a total of 537 Cd4+ T-cells from 11 mice remained, their ages being distributed as 143 (1 m), 122 (3 m), 67 (18 m), 107 (21 m) and 98 (30 m). Cells from any given age-group were derived from at least two mice (Supplementary Data 1), allowing us to take sample variability into account. Of the 537 Cd4+ T-cells, 186 were identified as being in the naïve state due to expression of *Lef1*, a well-known marker for naïve Cd4+ T-cells[22]. We used Seurat to perform feature selection, dimensional reduction and visualization (Methods), resulting in two broad clusters that correlated with age and Cd4+ T-cell subtype (Fig. 3a). Thus, this represents an "easy" scenario where ELVAR should be able to predict the expected shift to a more mature Cd4+ T-cell phenotype. As before, we ran ELVAR 100 times for each of 9 choices of the purity parameter value (range: $a = 0.1$ to $a = 0.9$), and using the same selection criteria as with our simulation model (Methods), we identified an optimal value of $a = 0.8$. We note that at this value, the number of inferred communities increased appreciably relative to Louvain, that purity was relatively close to the maximum, and that modularity remained relatively high (Fig. 3b). For each age group and for each of the 100 runs at this optimal $a = 0.8$ value, ELVAR communities enriched for cells from that age-group were identified (Methods, Fig. 3c, d). Ignoring sampling variability revealed a significant skew towards lower naïve cell-fractions in older mice (Fig. 3e). Taking mouse replicates into

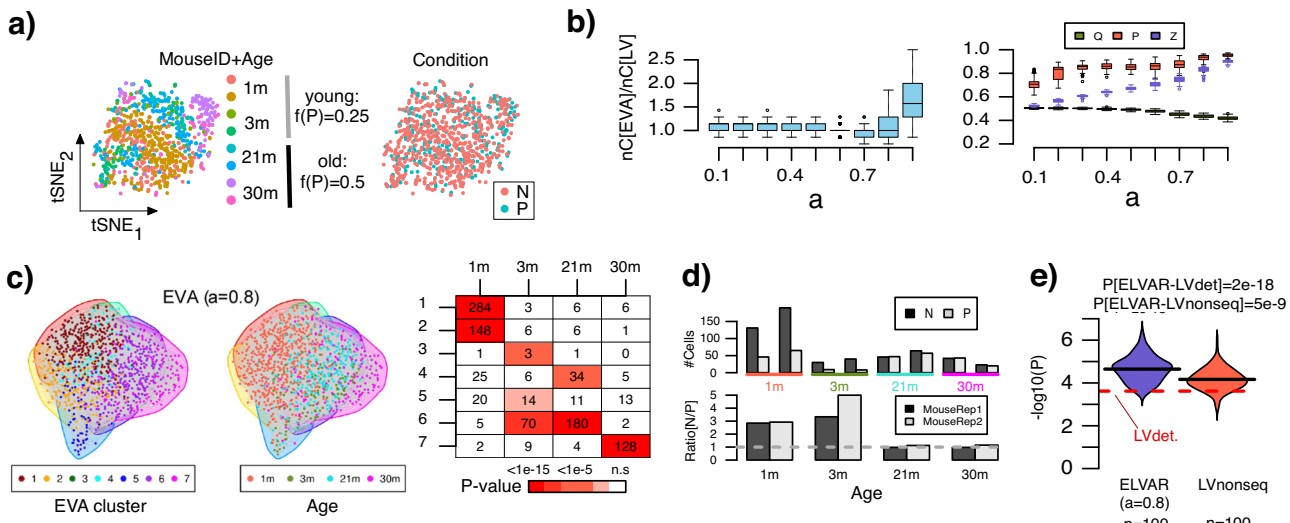

**Fig. 2 | Benchmarking EVA against Louvain. a** tSNE visualization of a simulated scRNA-Seq dataset consisting of 993 mouse cells drawn from 8 mice encompassing 4 age-groups (1 month, 3 months, 21 months and 30 months) with two mouse replicates per age-group. In the right panel, cells are annotated by perturbation state, where the frequency of cells being in the perturbed (P) state increases from 0.25 in young mice (1 & 3 m) to 0.5 in old mice (21 & 30 m). **b** Left: Boxplots displaying the ratio of the number of clusters inferred with EVA to the corresponding number inferred with Louvain (nC[EVA]/nC[LV], y-axis) for a range of different purity parameter values (x-axis). A total of $n = 100$ EVA runs were made at each purity parameter value. Right: Boxplots displaying the modularity (Q), purity (P) and generalized modularity (Z) as a function of purity index parameter $a$ for EVA. Each boxplot contains the values of 100 distinct EVA runs. Boxplot elements indicate median, interquartile range (IQR) and whiskers extend to 1.5 times the IQR. **c** Nearest neighbor cell-cell graph on which the EVA algorithm is run. Left panel: cells annotated by clusters inferred in one particular EVA run. Middle panel: cells annotated by age-group. Right panel: confusion matrix between the communities inferred with EVA (same run) and age-groups, with the number of cells and one-tailed Binomial test P-value of enrichment shown. Significance is assessed using Bonferroni adjustment at 0.05 level. **d** Barplot (top panel) displays the number of normal (N) and perturbed (P) cells from each mouse and age-group, using only cells from EVA communities enriched for specific age-groups (same run as in **c**). Barplot in lower panel displays the ratio N/P for each mouse replicate and age-group. **e** Violin plot compares the statistical significance (y-axis, -log₁₀P) of P-values from a negative binomial regression of perturbed cell number against age-group as derived from ELVAR (100 runs) against the corresponding statistical significance value derived from an analogous method that uses either the deterministic Louvain algorithm (LVdet) or a non-sequential (non-deterministic, 100 runs) version of Louvain (LVnonseq) in place of EVA. P-values shown are from a one-sided Wilcoxon rank sum test comparing the 100 ELVAR values to the LVdet one, or to the 100 values from LVnonseq. Source data are provided as a Source Data file.

account revealed the same skew independently of mouse-ID (Fig. 3f). To confirm this, we ran negative binomial regressions, which revealed highly significant and robust negative and positive associations for naïve and mature Cd4t cells, respectively (Fig. 3g). Hence, this confirms that the age-associated shift from naïve to mature Cd4t-cells is also present in lung tissue. Importantly, the associations of T-cell subfractions with age as obtained with ELVAR were significantly stronger than those obtained with the benchmark, i.e. with the analogous algorithm that uses the non-sequential Louvain algorithm in place of EVA (Fig. 3g, Methods).

Using the same lung-tissue data from the TMS[21], we also tested for age-associated differential abundance of alveolar macrophage M1/M2 polarization subtypes (Methods, Supplementary Data 2). A lower M1 to M2 ratio has been proposed to be a signature of lung cancer risk[24–26]. By applying ELVAR with age as the clustering attribute and macrophage polarization as the attribute of interest, we were able to detect a lower M1/M2 ratio with increased age (Supplementary Fig. S4), which was not evident when using the Louvain algorithm in place of EVA (Supplementary Fig. S4).

To demonstrate that ELVAR can detect disease relevant DA-shifts in other cell-types and biological conditions, we next considered two Covid-19 scRNA-Seq datasets[27,28]. Chua et al. profiled single cells in nasopharyngeal swabs from moderate and critically ill COVID-19 patients[27] (Methods). After QC, we retained ~13,500 immune cells encompassing 9 cell-types derived from 7 moderately and 11 critically ill patients (Methods, Supplementary Data 3). We applied ELVAR (100 runs) with disease severity as the clustering attribute to each of the 9 cell-types to determine if their abundance changes between moderate and severe Covid-19 cases. This confirmed an increased neutrophil and decreased monocyte-derived dendritic cell (moDC) counts in severe

cases (Supplementary Fig. S5). Using Louvain instead of EVA resulted in similar levels of statistical significance for neutrophils but less significant levels for moDCs (Supplementary Fig. S5). Finlay et al. profiled single-cells in the olfactory epithelium (OE) of Covid-19 patients who experienced long-term smell-loss (hyposmic) and those who did not (normosmic)[28]. After QC, we retained 11,173 relevant cells encompassing 5 cell-types (olfactory sensory neurons, sustentacular, Bowman glandular cells, microvillar and horizontal basal cells (HBCs)), derived from 5 hyposmic and 5 normosmic Covid-19 cases (Methods, Supplementary Data 4). We applied ELVAR (100 runs) to detect cell-types displaying DA in relation to the smell-loss phenotype, which was thus used as the clustering attribute. This revealed significantly decreased olfactory sensory neuron and increased microvillar counts in hyposmic cases (Supplementary Fig. S6a–c). However, in this dataset the degree of statistical significance attained by ELVAR was similar to the Louvain-benchmark (Supplementary Fig. S6c).

In summary, the analyses performed on these 4 scRNA-Seq datasets demonstrate (i) the ability of ELVAR to detect differential abundance patterns in relation to age and Covid-19 phenotypes, (ii) that in general ELVAR displays improved sensitivity over an analogous algorithm that uses Louvain in place of EVA, and thus (iii) that this improvement is solely due to the incorporation of cell-attribute information when inferring cellular communities.

## ELVAR predicts an increased stem-cell fraction in polyps from snRNA-seq data

We next applied ELVAR to a single-nucleus RNA-Seq (snRNA-Seq) dataset of colon cancer progression, encompassing normal samples from healthy individuals (N), normal samples from unaffected familial adenomatous polyposis (FAP) cases (A), polyps from predominantly

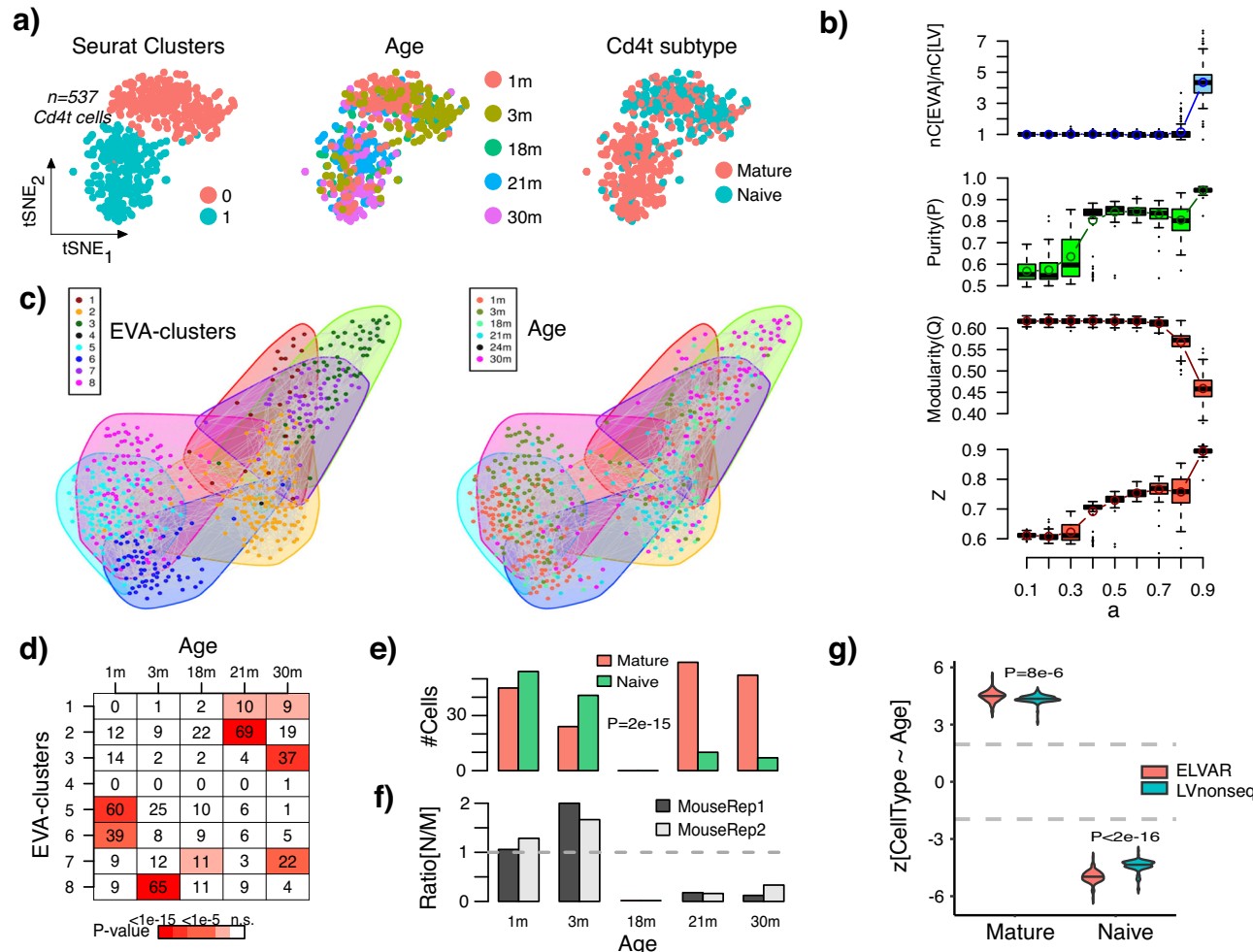

**Fig. 3 | Validation and benchmarking of ELVAR in Cd4t cells from lung tissue.**
**a** tSNE visualization of 537 Cd4+ T-cells with cells annotated by inferred Seurat cluster (left), by age-group (middle) and by Cd4+ T-cell subset (naïve vs mature) (right). **b** Top panel: boxplots display the number of communities inferred using EVA against the purity parameter *a*, normalized relative to the number of communities inferred with the Louvain algorithm (*a* = 0). Each boxplot represents the distribution over 100 different runs. Middle panel: As top-panel but now with y-axis displaying the purity of the clusterings. Lower panel: As top-panel but now with y-axis displaying the modularity of the clusterings. Boxplot elements indicate median, interquartile range (IQR) and whiskers extend to 1.5 times the IQR. **c** Left panel: Cell-cell nearest neighbor graph inferred using Seurat, with cell colors indicating the inferred EVA communities from one typical run. Right panel: as left-panel, but with cells now colored by age-group. **d** Matrix entries give the number of cells per EVA-cluster and age-group, with color indicating the *P*-value of enrichment

from a one-tailed Binomial test, for one particular run. For a given age-group, only cells from enriched clusters are taken forward using a Bonferroni-adjusted threshold (typically around 0.001). **e** Barplot displaying the number of mature and naïve Cd4t cells per age-group only using enriched clusters from **d**. *P*-value is from a two-tailed Fisher-test. **f** Barplots display for each mouse-replicate the ratio of naïve to mature Cd4t cells. **g** Violin plots displaying the z-statistics (*n* = 100 runs, i.e each violin contains 100 datapoints) derived from the negative binomial regression for the case of naïve and mature cell-type fractions. The gray dashed lines indicate the level of statistical significance (*P* = 0.05). The violin plots and displayed *P*-values compare the z-statistics of ELVAR to the corresponding z-statistics derived from using the non-sequential Louvain algorithm (LVnonseq) in place of EVA. *P*-values derive from a one-tailed Wilcoxon rank sum test. Source data are provided as a Source Data file.

FAP cases (P) and colorectal cancer adenomas (A), encompassing over 200,000 cells[29]. We asked if ELVAR could detect cancer-associated DA-shifts in the stem-cell and T-regulatory cell populations, because in the original study by Becker et al.[29] an increase in the epithelial stem-cell and regulatory T-cell fractions was only observed when analyzing scATAC-Seq data, and not when analyzing the snRNA-Seq data itself which displayed very high (97–99%) sparsity. We reasoned that ELVAR's improved sensitivity would allow detecting these shifts from the snRNA-Seq data itself. To ensure robustness, we performed the analysis in two independent ways (Methods). In the first approach, we restricted to a subset of samples for which the QC processing and cell-type annotation was already provided in the original study[29] (Supplementary Data 5-6). Applying ELVAR (100 runs) to the cell-cell similarity graphs with disease stage as the clustering attribute, derived separately for enterocytes and lymphocytes, revealed statistically

significant progressive increases in the stem-cell and regulatory T-cell fractions, despite the relatively small numbers of samples (Fig. 4). Of note, the statistical significance levels of these DA-shifts were much stronger for ELVAR compared to the analogous method that uses Louvain instead of EVA (Fig. 4f). In fact, when using Louvain instead of EVA we did not observe a clear increase of regulatory T-cells, further attesting to the improved sensitivity of a cell-attribute aware clustering method.

In the second approach, and with the aim to increase sample size, we re-analyzed the full snRNA-Seq dataset, performing QC and re-annotating cells into broad enterocyte, goblet, immune-cell, stromal and endothelial cell categories (Methods). Briefly, to annotate, we identified broad cell-types using only normal samples and well-known cell-type specific markers, to subsequently build an mRNA expression reference matrix, which was then used in a robust partial correlation

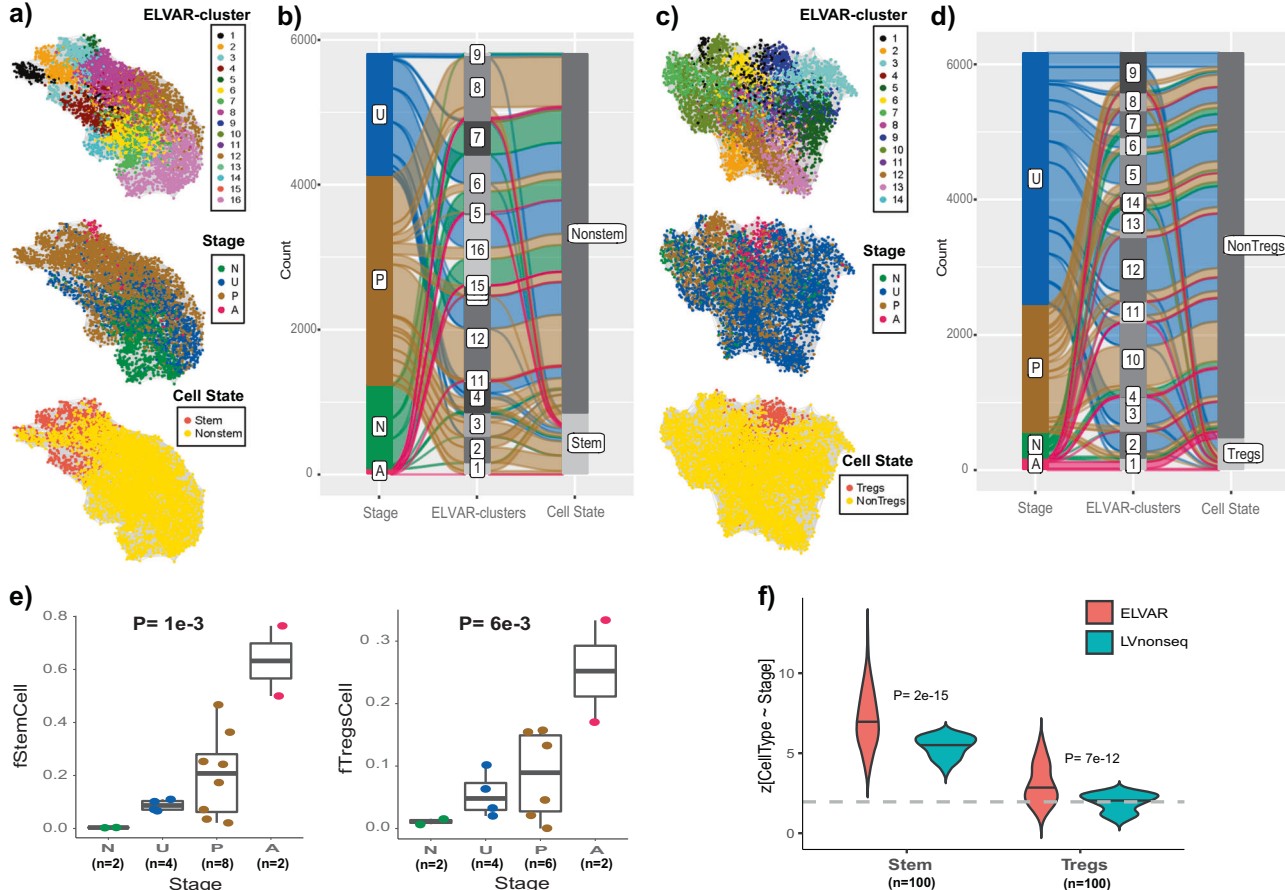

**Fig. 4 | ELVAR predicts increased stem-cell and T-regulatory cell fractions in polyps. a** Top panel: The cell-cell similarity graph inferred using Seurat on scRNA-Seq data with epithelial enterocyte lineage cells annotated by community membership, as inferred using ELVAR. Middle and lower panels depict the same graph but with cells annotated by disease stage (N = normal, U = unaffected, P = polyp, A = adenoma) and cell-state. Data is shown for one representative ELVAR run. **b** Alluvial plot displaying composition of ELVAR communities according to disease stage and cell-state. **c** As **a** but for lymphocyte-cells in colon tissue. **d** As **b** but for the lymphocyte cells in colon-tissue. **e** Boxplots displaying the stem-cell (left panel) and T-regulatory cell (right panel) fraction as a function of disease stage, considering only cells that are part of significantly enriched ELVAR-clusters. *P*-value derives from a two-sided linear regression. Boxplot elements indicate median, interquartile range (IQR) and whiskers extend to 1.5 times the IQR. **f** Violin plots comparing ELVAR to the analogous DA-testing algorithm that uses non-sequential Louvain in place of EVA ("LVnonseq"). The y-axis labels the corresponding z-statistics from negative binomial regressions (100 runs each) of stem-cell or T-regulatory cell counts against disease stage. Gray dashed line indicates the *P* = 0.05 significance level. *P*-values shown derive from a one-tailed Wilcoxon rank sum test comparing the ELVAR z-statistic distribution to the one derived using "LVnonseq". Source data are provided as a Source Data file.

framework[30,31] to annotate all cells from all disease stages (Methods). Only cells that were confidently annotated into one of the broad categories were taken forward for further analysis (Methods). Whilst the high sparsity of the snRNA-Seq data precluded reliable annotation of T-regulatory cells, in the case of stem-like cells, we applied a recently validated single-cell transcription-factor (TF) regulon-based method called CancerStemID[32], that first estimates differentiation activity of colon-specific transcription factors (TFs)[33] across all cells, subsequently identifying stem-like cells as those displaying the lowest average differentiation activity (Methods). We observed that the average differentiation activity of the colon-specific TFs decreased during cancer progression (Supplementary Fig. S7a). In total, we identified 38,667 stem-like and 65,432 non-stem cells, with the stem-like cells displaying much lower levels of differentiation activity (Supplementary Fig. S7a). Next, we applied ELVAR (100 runs) with disease stage as the clustering attribute (Supplementary Fig. S7b). Alluvial plots indicated that inferred communities enriched for disease stages were also predominantly associated with either stem or non-stem cells (Supplementary Fig. S7c). ELVAR confirmed an increase in the stem-cell fraction, which was particularly pronounced at the polyp-stage (Supplementary Fig. S7d), and which was further validated using NBRs to account for inter-subject variability (Supplementary Fig. S7e).

In summary, these results demonstrate that ELVAR is able to detect shifts in relevant cell-states from snRNA-Seq data, thus extending and confirming earlier findings derived from scATAC-Seq data[29].

## ELVAR compares favorably to non-clustering based DA-testing methods

Having demonstrated that ELVAR can successfully detect DA of various cell-types in different biological contexts and that the cell-attribute aware clustering step improves the sensitivity of the procedure, we next compared ELVAR to two competing non-clustering based methods called DA-seq[7] and Milo[10]. Of note, although DA-seq and Milo allow DA to be assessed in relation to one main cell-attribute, they do not explicitly allow assessment of DA of additional cell-attributes (e.g. cell-types) relative to the main one. Thus, in order to compare ELVAR to DA-seq and Milo in their ability to detect DA of cell-types in relation to a biological condition such as age or disease stage, we adapted the DA-seq and Milo algorithms to this particular DA-task (Methods). We applied these two methods in the context of all previously analyzed datasets including the lung-tissue Cd4t-cell and alveolar macrophage TMS scRNA-Seq data, the two Covid-19 related scRNA-Seq sets and the colon enterocyte snRNA-Seq dataset. ELVAR attained stronger levels of statistical significance compared to DA-seq or Milo (Fig. 5a, c, e, g, i).

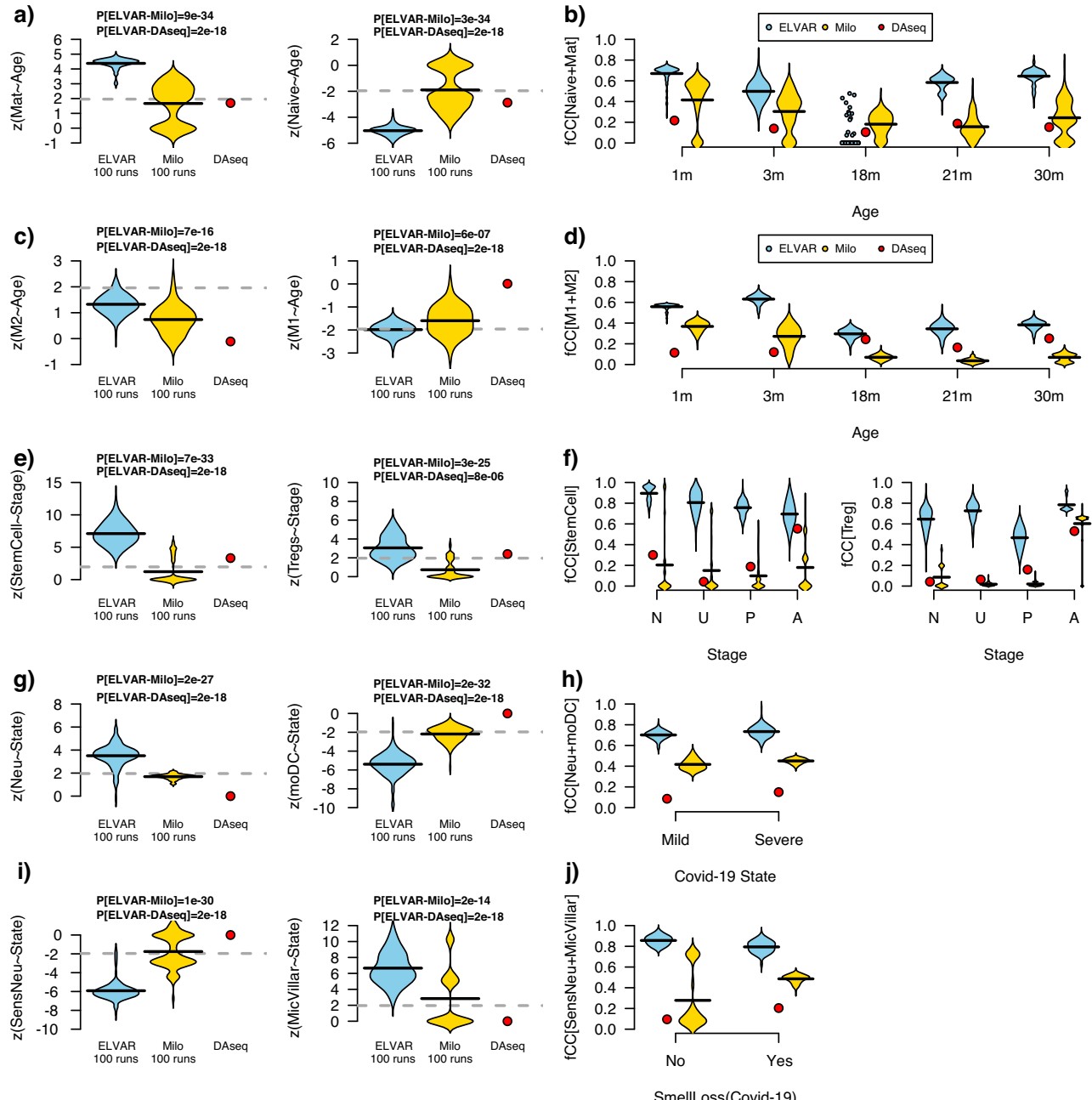

**Fig. 5 | ELVAR compares favorably to DA-seq and Milo. a** Violin plots displaying the z-statistics of associations between cell-counts (y-axis, mature Cd4t-cells and naïve Cd4t-cells) and age derived from negative binomial regressions for ELVAR, Milo and DAseq, as shown. Gray dashed line indicates the threshold of statistical significance *P* = 0.05. *P*-values derive from a one-tailed Wilcoxon rank sum test comparing ELVAR derived z-statistics to those from DA-seq and Milo, respectively. **b** Violin plots displaying the fraction of captured mature and naïve Cd4t cells (fCC[Naïve+Mat], y-axis) for each biological condition (x-axis) and method. **c, d** As **a, b** but for alveolar macrophage M1 and M2 subtypes. **e, f** As **a, b**, but for stem-cell and T-regulatory cell fractions in colon tissue in relation to colon cancer progression (N = normal, U = unaffected FAP cases, P = polyp, A = adenoma). **g, h** As **a, b** but for neutrophil (Neu) and monocyte-derived dendritic cells (moDC) fractions in relation to Covid-19 disease severity. **i, j** As **a, b** but for sensory neurons (SensNeu) and microvillar cells (MicVillar) in relation to Covid-19 smell-loss phenotype. Source data are provided as a Source Data file.

For instance, whilst all 3 methods correctly predicted an age-related decrease of naive Cd4t cells, DA-seq and Milo only attained marginal levels of significance, in contrast to the much stronger levels of statistical significance obtained with ELVAR (Fig. 5a). More strikingly, the increased stem-cell and T-regulatory fractions with colon cancer progression was not evident at all when using Milo and only marginally so when using DA-seq (Fig. 5e). Likewise, in the Covid-19 datasets, the associations were much stronger for ELVAR, marginally significant for Milo, whilst DA-seq did not achieve significance (Fig. 5g–i).

We reasoned that the improved sensitivity of ELVAR may be related to its ability to capture larger communities specifically enriched for cells representing the various biological conditions. To test this, we computed the fraction of captured cells for each biological condition, defined as cells of a given condition that belong to communities (ELVAR) or cellular neighborhoods/regions (Milo/DA-seq) significantly enriched for that condition (Methods). Supporting our hypothesis, we observed that ELVAR captured significantly more cells from each biological condition compared to Milo or DA-seq

(Fig. 5b, d, f, h, j). It is noteworthy for instance, that in the case of Cd4t-cells, ELVAR's improvement over Milo and DA-seq was specially pronounced for the oldest age-groups (30 m), whilst there was no improvement for the intermediate group (18 m). This supports the view that Milo and DA-seq struggle to capture larger communities of old cells, probably due to these cells displaying higher heterogeneity and therefore less prone to cluster together in local neighborhoods. In contrast, by incorporating age as a cell-attribute when inferring communities, ELVAR is able to extend its' influence beyond the local neighborhoods to capture larger clusters of old cells, thus improving power and facilitating the detection of age-related shifts in underlying cell-states. We note once again that although this improvement in power over DAseq and Milo comes at the expense of increased runtimes when compared to Milo, that ELVAR runtimes remain very feasible (Supplementary Fig. S3).

**ELVAR is robust to batch effects and false positives in cell-type annotation**

Finally, we assessed ELVAR's performance in relation to batch effects and false positives in cell-type annotation. We reasoned that in the context of DA-testing, sample batch correction may not always improve the signal-to-noise ratio (SNR) because sample is correlated with the biological condition of interest. We further reasoned that ELVAR, by virtue of using the biological condition as the clustering attribute, can help circumvent batch effects by drawing-in together cells from different samples/batches but same biological condition. Using the previously analyzed datasets, we thus compared ELVAR's DA-test z-statistics for three different scenarios: (i) no batch correction, (ii) prior batch-correction with Harmony[34] and (iii) prior batch correction with Seurat[19]. In general, we observed that ELVAR's performance was relatively robust to whether prior batch correction is performed (Fig. 6a–d). Importantly, the degree of statistical significance was generally speaking higher without prior batch correction, although this was study-specific (Fig. 6a–d). In fact, for those datasets and cell-types (e.g. sensory neurons in Fig. 6c) where prior batch correction improved significance levels, the improvement itself did in general not alter the sensitivity to detect significant changes (Fig. 6a–d). This is because in these scenarios the DA-shifts were substantial in terms of effect size. Hence, these results confirm the view that in cases where DA-shifts are of smaller effect size, batch correction may further reduce the SNR, and hence that using ELVAR without prior batch correction may improve power.

Since cell-type annotation is an error-prone procedure, we also assessed ELVAR's robustness to an increased false positive rate (FPR) when annotating cells to specific cell-types. We thus simulated an increased FPR in each of the previous four datasets (Methods), recording the sensitivity to detect a significant DA-shift for the relevant cell-types across a total of 100 distinct Monte-Carlo runs. We note that errors in cell-type annotation only affect the NBR-step of ELVAR since cell-type annotation itself is not used when inferring cellular communities. Consequently, the sensitivity to detect the DA-shifts remained consistently high in each study until the FPR reached a study-specific threshold, at which point sensitivity dropped markedly (Fig. 6e). In general, these thresholds on the FPR in cell-type annotation were quite high indicating that under reasonable FPRs ELVAR's performance is robust.

In principle, false positives can also arise due to errors in diagnosis or staging of a disease, which would introduce errors in the clustering attribute and thus affect ELVAR's performance. Although unlikely, such errors in diagnosis could be present in the Covid-19 and colon-polyp datasets analyzed here. By definition, such errors in diagnosis or staging affects all cells of a sample. Hence to simulate such errors we randomly flipped a small percentage (~20%) of sample phenotype labels, subsequently rerunning ELVAR to test its robustness to detect the same cell-type shifts as in the unperturbed scenario (Methods). We

observed that ELVAR's sensitivity to detect DA-shifts typically dropped by about 20%, although for some cell-types the drop was significantly less, for instance, sensitivity dropped by only 5% when detecting an increase of the enterocyte stem-cell fraction with colon-cancer progression. On the other hand, the sensitivity to detect a decrease of sensory neurons in the OE of Covid-19 patients experiencing long-term smell-loss, dropped by as much as 30%, to remain at just over 60% (Supplementary Fig. S8). Thus, even at a relatively high FPR of ~20%, sensitivities to detect DA-shifts for all cell-types remained at over 60%. This indicates that ELVAR is relatively robust to such errors, although achieving high sensitivity clearly hinges on the FPR in the clustering attribute being reasonably low.

## Discussion

Recent studies have advocated DA-testing methods (e.g. Milo, DA-seq) that infer cellular states and associated DA-patterns from enriched cellular neighborhoods or regions within the high-dimensional single-cell state manifold[7,10]. These studies have argued that since DA-patterns are often sought for cell-states that do not cluster well in the manifold, that hard clustering algorithms such as Louvain are inappropriate tools for inferring the cellular states and their underlying DA-patterns. Instead, fuzzier representations of communities, known as local neighborhoods/regions, are better suited for the DA-task. Here we have shown that an alternative solution is to use cell-attribute information when clustering cells. By using cell-attribute information in the community inference procedure, one can more readily discern cellular communities defined by the biological condition (i.e. the clustering attribute) itself, thus helping to circumvent the noise and orthogonal sources of variation which would otherwise preclude identification of such states. Furthermore, in the applications to aging, Covid-19 and colon cancer progression considered here, ELVAR displayed higher sensitivity than Milo and DA-seq to detect biologically important and plausible DA-shifts, such as the age-related shift from naïve to mature Cd4+ T-cells in lung tissue, thus mirroring the corresponding known shift in blood[22,24], or the increased epithelial stem-cell and T-regulatory fraction in polyps. We also contributed a theoretical understanding underpinning this improved sensitivity, as demonstrated by ELVAR's ability to detect cellular communities enriched with larger numbers of cells belonging to specific biological conditions. For instance, ELVAR enabled the identification of communities representing immune-cell states in old cells, which competing methods like Milo or DA-seq could not resolve due to increased heterogeneity of the older cells. Indeed, we stress that due to the cell-attribute-aware clustering, ELVAR was able to capture more cells of a given biological condition within a community of cells enriched for that biological condition, compared to neighborhood approaches like Milo/DA-seq, thus increasing power to detect subtle DA-shifts. In this regard, it would be interesting to explore if the improved sensitivity would also be seen relative to miloDE[35], a recently proposed extension of Milo, which uses a 2nd order k-nearest neighbor graph approach to generate a state manifold that displays more homogenous neighborhoods. Indeed, miloDE has been shown to significantly increase power in downstream differential expression tasks.

ELVAR's improved sensitivity to detect DA-shifts was also seen when benchmarked against an analogous clustering-based DA-method that uses Louvain in place of EVA. This benchmarking is important as it disentangles the effect of using a cell-attribute when clustering from one that does not, thus highlighting the specific importance of using cell-attribute aware clustering. Of note, the improvement of ELVAR over its Louvain-analog was more pronounced in studies displaying weaker clustering structure (e.g. the immune-cell subsets changing with age in lung tissue, or the stem-cell state increasing in polyps), whilst the improvement was much less noticeable in those studies with stronger clustering (e.g. neutrophils changing with Covid-19 disease severity). This supports the view that cell-attribute aware clustering

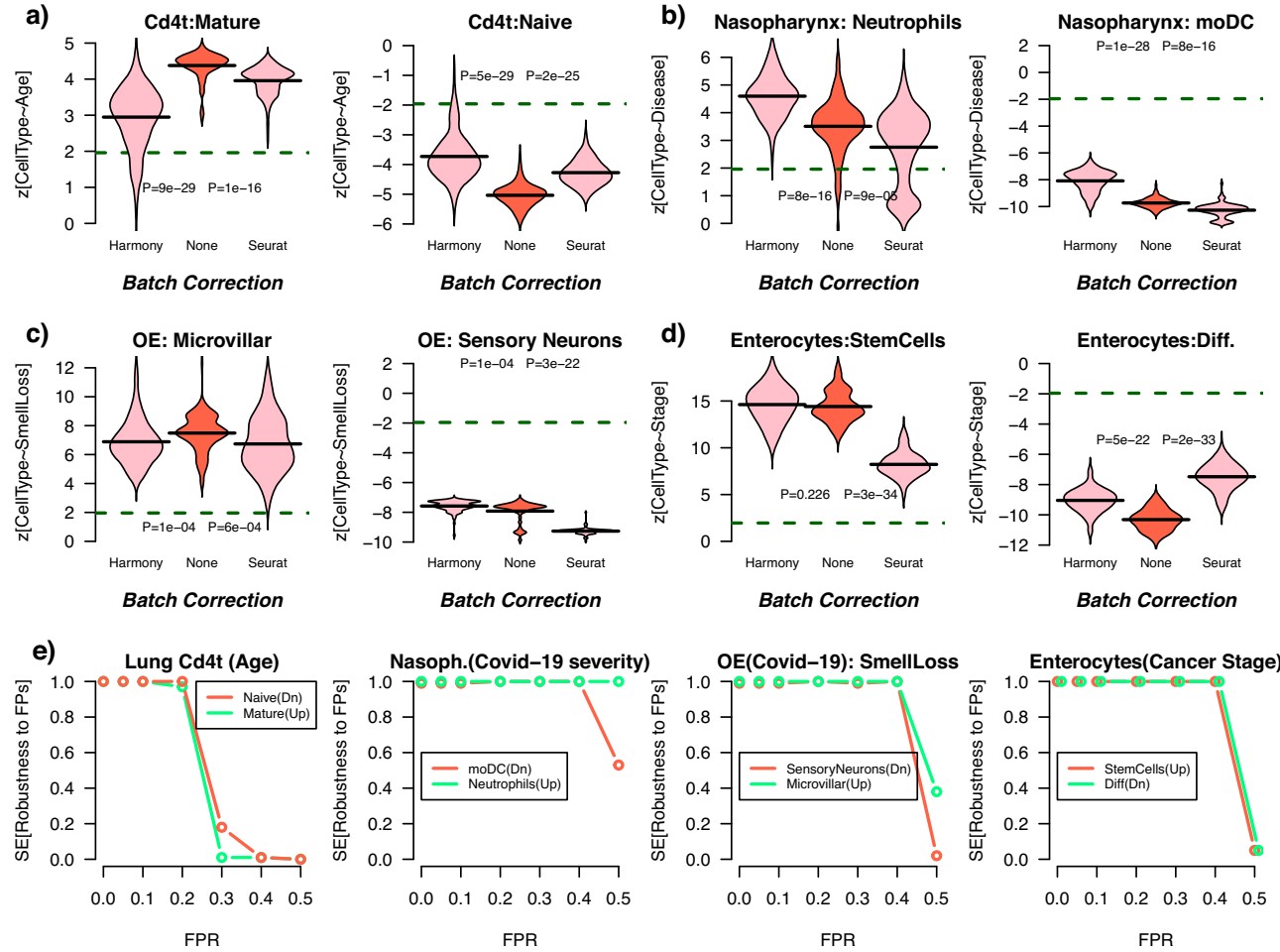

**Fig. 6 | ELVAR is robust to batch correction and false positives in cell-annotation. a** Violin plots of negative binomial regression z-statistics of association of cell-type counts with age for mature and naïve lung-tissue Cd4t cells and for 3 different scenarios: None = ELVAR was run with no batch (sample) correction, Harmony = ELVAR was run data batch-corrected with Harmony, Seurat: ELVAR was run on data batch-corrected with Seurat. Each violin plot contains the values for 100 distinct ELVAR runs. The *P*-values are derived from two-tailed Wilcoxon rank sum tests comparing "None" to either "Harmony" or "Seurat". Horizontal dashed lines indicate the *P* = 0.05 significance level. **b** As **a** but for nasopharyngeal neutrophils and monocyte-derived dendritic cells (moDC) cell counts changing with Covid-19 disease severity (mild vs critical). **c** As **a** but for microvillar and sensory olfactory neurons in the olfactory epithelium (OE) in relation to their counts changing with long-term smell loss in Covid-19 patients. **d** As **a** but for colon

enterocyte stem-cell and differentiated cell counts changing with cancer stage progression. **e** Left panel: Sensitivity to detect a significant (NBR *P*-value < 0.05) change in the abundance of naïve and mature Cd4T-cells with age in the lung tissue of mice under different rates of false positives (FPR, x-axis). Sensitivity was estimated over 100 distinct runs. Middle left panel: As left panel, but for detecting a significant change in the abundance of neutrophils and monocyte-derived dendritic cells (moDC) with Covid-19 disease severity. Middle right panel: As other panels, but for detecting a significant change in the abundance of olfactory sensory neurons and microvillar cells with Covid-19 smell-loss phenotype. Right panel: as other panels, but for detecting a significant change in the abundance of enterocyte stem and differentiated cells with colorectal adenoma progression. Source data are provided as a Source Data file.

can improve the identification of relevant cellular states, which otherwise can't be discerned due to the noise in scRNA-Seq data.

We have also seen how ELVAR can circumvent the need for batch correction, as performing batch-correction prior to ELVAR rarely led to an improvement in sensitivity. Of note, whilst ELVAR also displayed significant robustness to FPs in cell-type annotation, it was more sensitive to putative FPs in the clustering attribute representing the sample/phenotype label. This is not unexpected, since although the rate of FPs in cell-type annotation could be relatively high, the cell-type annotation itself is not used in the clustering inference step. On the other hand, FPs in the clustering attribute can affect the quality of the inferred communities. It is worth pointing out though that in practice FPRs in sample annotation are generally much lower than the highly conservative 20% error-rate considered here.

The improved inference of DA with cell-attribute aware community detection inevitably comes at the expense of increased computational complexity and runtimes compared to Milo or the

Louvain-analog. However, runtimes remain very feasible and hence the increased computational complexity does not present a practical limitation. Indeed, a typical scRNA-Seq study may profile on the order of 200k cells, encompassing on the order of 10 cell-types, hence on the order of 20k cells per cell-type. Since ELVAR is aimed at detecting subtle shifts of underlying cell-states within one of these cell-types (e.g. mature vs naïve Cd4+ T-cell states), the typical cell-cell graphs on which we would apply ELVAR would have on the order of 20k cells. On such a network, one ELVAR run is completed in ~10–15 min. On cell-cell networks encompassing ~100k cells, one ELVAR run would complete in the order of 80–120 min. Multiple runs can be easily parallelized, for instance, to identify an optimal *a* parameter value may require a total of 100 runs for each of nine *a* parameter values, so a total of 900 runs, which on a 100-node server would require 9 instances. Thus, for a typical 20k cells per cell-type, the total runtime to complete the whole ELVAR-task on a 100-node server would be ~90–135 min.

A nice feature of the ELVAR algorithm is that the resolution (i.e. average cluster size) of the inferred communities is dependent on the same parameter that controls the purity of the communities. As shown here, the optimal purity parameter value is generally in the range 0.7–0.9. Whilst this appears to be close to the value 1, i.e. the value at which modularity ceases to enter the objective function, it is worth noting that purity and modularity are relatively stable for all $a$ parameter values less than 0.7 and that deviations from the Louvain-benchmark are only seen when $a$ is at least 0.7. Thus, from a practical perspective, purity does not dominate the clustering until it is very close to 1. This can be understood by the fact that purity plays no role in the construction of the cell-cell nearest neighbor graph, the latter being solely determined by the scRNA-Seq profiles. In other words, the topological structure of the cell-cell graph, which is determined purely by the scRNA-Seq profiles, limits the way purity can influence clustering solutions, even when purity values are close to 1.

It is also important to re-emphasize the need to infer cellular states within the context of the high-dimensional single-cell state manifold. Given two cellular attributes that are defined to a large extent independently of this high-dimensional manifold, one can in principle always perform DA-testing between these two attributes using NBRs or another statistical framework, without the need to apply an algorithm to the state-manifold[11]. For instance, in the case of age and a cell-state defined by the binary expression of a single marker gene, DA-testing with multiple replicates could be done using NBRs on cell-counts within this two-dimensional attribute space (age, cell-state). However, in the context of scRNA-Seq data, such an approach has been shown to be suboptimal[10], because it assumes, unreasonably so, that all cells sharing common attributes define the same cellular states. Thus, a cell-attribute aware clustering and DA-testing pipeline such as ELVAR strikes an optimal balance, allowing more biologically relevant and robust cell-states to be inferred, whilst simultaneously also removing the many noisy and rogue cells that are not part of these states.

Finally, we stress the importance of developing sensitive methods for DA-testing. As shown here, by applying ELVAR to snRNA-Seq data we were able to predict increased stem-cell and T-regulatory fractions in polyps preceding colorectal adenoma, when such DA-patterns were previously only observed using scATAC-Seq data[29]. Thus, this finding has important repercussions for the biomedical field in demonstrating that snRNA-Seq data is perfectly adequate to detect DA-changes that are likely to be informative of disease risk.

In summary, ELVAR, and the cell-attribute aware clustering algorithm on which it is based, is a useful addition to the arsenal of statistical methods for DA-testing in scRNA-Seq and snRNA-Seq data. Given the richness and complexity of single-cell omic data, including multi-omic data, general network science approaches will continue to find successful applications in this area.

## Methods

### Single-cell RNA-Seq datasets
We here analyzed the following scRNA-Seq datasets:

**Tabula Muris Senis (TMS).** This mouse scRNA-Seq dataset[21] encompasses many different tissue-types with samples collected at 6 different ages: 1, 3, 18, 21, 24 and 30 months. Data object files were downloaded from figshare https://doi.org/10.6084/m9.figshare.8273102.v2.

We used the normalized data as provided in the h5ad files. We focused on the lung-tissue 10X dataset, because it contained one of the largest numbers of immune-cell subtypes with good representation across age-groups including multiple mouse replicates.

**Colon cancer development.** This is a human snRNA-Seq dataset[29] encompassing colon samples collected from healthy individuals, normal samples from unaffected individuals with FAP, polyps from FAP

and non-FAP cases, and colorectal adenomas. We analyzed both the processed snRNA-seq data available from GitHub (https://github.com/winstonbecker/scCRC_continuum), as well as the full unprocessed data available from GEO (GSE201348). Processed data were stored as Seurat objects that included donor, disease stage and cell-type annotation information.

### The EVA algorithm
Here we describe the recently published algorithm (EVA: Extended LouVain Algorithm) to identify homogeneous communities in a network with node attributes[17]. Let $G = (V, E, A)$ denote a graph where $V$, $E$ and $A$ are the set of vertices, edges and node (cell) attributes, respectively. Node/cell attributes can be categorical or numerical such that $A(v)$, with $v \in V$, identifies the set of cell attribute values associated with cell $v$. In our applications, we will mostly consider one cell attribute (typically the cell's age or perturbation state) but below we formulate the model for any number of node attributes. With EVA, the goal is to identify a network partition, i.e. a mutually exclusive set of communities/clusters, that maximizes a topological clustering criterion as well as node label homogeneity within each community. Thus, the measure we wish to maximize consists of two components: the modularity $Q$ that measures the extent to which the partitioning captures clusters of high-edge density, and a purity index $P$ that measures the homogeneity of the communities in relation to node attributes.

In more detail, the modularity $Q$ of a partition quantifies the edge density within communities relative to that expected under an appropriate null distribution[36], and is defined by

$$Q = \frac{1}{2m} \sum_{vw} \left[ A_{vw} - \gamma \frac{k_v k_w}{2m} \right] \delta(c_v, c_w) \tag{1}$$

where $m$ is the number of edges, $A_{vw}$ represents the edge weight between nodes $v$ and $w$, $k_v = \sum_w A_{vw}$ and $k_w = \sum_v A_{vw}$ are the sum of weights of the edges linked to nodes $v$ and $w$, respectively. The Kronecker $\delta(c_v, c_w)$ function is 1 when nodes $v$ and $w$ are in the same community ($c$) and 0 otherwise. The resolution parameter $\gamma$, which typically takes values in the range $0 < \gamma \leq 1$, controls the number and size of inferred communities with higher resolution values leading to a greater number of smaller communities. For $\gamma = 1$, $Q$ can take values between 0 and 0.5[17].

The purity index is defined by the average node label homogeneity (i.e. purity) over all inferred communities:[17]

$$P = \frac{1}{|c|} \sum_{c \in C} P_c \tag{2}$$

where $P_c$ represents the purity of community $c$. The purity of a given community ($c$) is defined as the product of frequencies of the most frequent node attribute values within the community $c$

$$P_c = \prod_A \frac{\max_a(\sum_{v \in c} a(v))}{|c|} \tag{3}$$

where $A$ is the set of node attributes, and $a(v)$ is an indicator function for node $v$ taking value 1 if $a = A(v)$, 0 otherwise. $P_c$ attains maximum purity 1 when all nodes in the community ($c$) have same attribute value sequence. In the case of just one cell attribute, this corresponds to the case where all cells in the community have the same attribute value. Of note, $P$ takes values in the range 0 to 1.

Finally, the EVA algorithm is defined by the optimization of a generalized modularity function $Z$

$$Z = \alpha P + (1 - \alpha) Q \tag{4}$$

where $\alpha$ is the purity index parameter taking values also in the range 0 to 1. Of note, for $\alpha = 0$, $Z = Q$ and we recover the Louvain algorithm if $\gamma = 1$[18] (or a modified Louvain algorithm if $\gamma < 1$). At the other extreme ($\alpha = 1$), $Z = P$, and the clustering algorithm only cares about maximizing purity subject to network connectivity constraints. For a given $\alpha$ and $\gamma$, we optimize $Z$ following the algorithmic implementation of Citraro and Rossetti[17].

## The ELVAR algorithm

Building on EVA, we developed an algorithm and R-package called ELVAR for differential abundance (DA) testing in scRNA-Seq data. Specifically, the biological question being addressed by ELVAR is whether the proportion of a given cell-state or cell subtype (the attribute of interest) changes in relation to some other factor or cell attribute such as age or disease stage (the clustering attribute). Given the potentially high technical and biological variability, such DA-testing should ideally be carried out in scenarios where multiple replicates are available[10]. ELVAR is designed for complex scenarios where (i) the source of variation associated with the biological conditions, cellular states or subtypes is relatively small and (ii) where sample replicates are available. The ELVAR algorithm consists of 4 main steps that we now describe:

(i) *Construction of the k nearest-neighbor (knn) cell-cell graph:* As input, the EVA algorithm requires a connected nearest neighbor cell-cell graph, as generated for instance using Seurat's *FindNeighbors* function. The number of nearest neighbors $k$ should be chosen sensibly in relation to the total number of cells $n_{tot}$. For instance, we aim for a ratio $n_{tot}/k \sim 50$, so that for a 1000 cell graph, the number of neighbors is 20. To run EVA, the input graph must also have at least one vertex/node (i.e. cell) attribute, which will be used when inferring communities in the graph. In our applications this clustering attribute will be age or disease status. In general, cells also have other attributes besides the one being used in the community-inference process. For instance, these additional attributes could be the sample replicate (individual/mouse) from which the cell derives, or a particular cell-state or subtype. To reiterate and avoid confusion we call the attribute used in the clustering or community inference as the "clustering attribute", whilst the attribute being interrogated in DA-testing as the "attribute of interest".

(ii) *Selection of purity index parameter a:* Another important input to the EVA algorithm is the value of the purity index parameter $a$ (called $\alpha$ in previous section), which controls the relative importance of purity $P$ over modularity $Q$ when optimizing $Z$. Typically, we recommend running the EVA algorithm 100 times for different $a$ parameter values ranging from 0.1 to 0.9 (the extremes $a = 0$ and $a = 1$ are not interesting), in order to assess how $P$, $Q$ and $Z$ vary as a function of $a$. It is also important to record the number of inferred communities, as this also depends on the value of $a$. Following the recommendation of Citraro et al.[17], we choose $a$ such that when compared to Louvain ($a = 0$) there is a clear increase in the number of inferred communities. However, we also modify their criterion by taking the purity $P$ and modularity $Q$ into consideration. Specifically, we seek an $a$-value which not only leads to a significant increase in the number of inferred clusters, but which also achieves a relatively high purity without much degradation in the modularity $Q$, all measured relative to the Louvain solution. More quantitatively, as far as the relative number of inferred clusters is concerned, we choose all $a$-values for which the 95% quantile of cluster number ratios (ratio taken relative to Louvain and quantile is taken over the 100 runs) is greater or equal than 1.5. As far as the purity is concerned, we select all $a$-values for which the mean purity taken over the 100 runs is at least 75% of the maximum purity value measured at $a = 0.9$. Finally, as far as the modularity is concerned, we choose all

$a$-values for which the mean modularity taken over the 100 runs is at least 75% of the maximum modularity measured at $a = 0.1$. To arrive at a final optimal $a$-value we then take the intersection of the three sets of permissive $a$-values, which generally leads to a unique $a$ parameter value satisfying all three criteria above. The reason why these criteria generally lead to a unique $a$-value is that the purity and cluster number criteria generally select the larger $a$-values down to a minimum, whereas the modularity criterion generally selects the smaller $a$-values up to a maximum. In there is no overlap, the three thresholds (95% quantile, ratio 1.5, 75% of maximum purity and modularity) can be altered to ensure a common $a$-value. Of note, although for a fixed $a$ EVA works by optimizing $Z$, we do not choose the $a$ value which maximizes $Z$, because for most real-world networks, $Z$ will increase with $a$ and will be maximal when $a = 1$. For typical *knn* cell-cell networks, the optimal $a$ value is generally in the range 0.7 to 0.9. As far as the other resolution parameter is concerned, we generally consider $\gamma = 1$, as this allows direct benchmarking to the original Louvain algorithm which is also defined for $\gamma = 1$.

(iii) *Inference of enriched communities with EVA:* Having inferred the optimal parameter $a$ value, we now rerun EVA with this input parameter $a$ value, to infer communities. The next step is to then identify those communities that are enriched for specific clustering attribute values. This is done for each clustering attribute value in turn, using a Binomial test with a stringent Bonferroni-adjusted $P$-value $< 0.05/$(number of communities * number of clustering attribute values) threshold. Only communities enriched for specific clustering attribute values are taken forward for further analysis. Thus, the purpose of this important step is to remove cells that don't clearly define cell-states associated with the clustering attribute value. Having found all enriched communities for a given clustering attribute value, we then group together all cells from these enriched communities to define a cell-group per clustering attribute value. The cells within a cell-group may derive from distinct sample replicates.

(iv) *DA testing for an attribute of interest:* For each cell group associated with a clustering attribute value $v$, we next count the number of cells with any given attribute-of-interest value contributed by each replicate $r$, whilst also recording the total number of cells contributed by that sample replicate. For each attribute-of-interest value, we then run a negative binomial regression (NBR) of the cell count against the clustering attribute value (here assumed ordinal e.g. age-group or disease stage) with the sample replicate's total cell count being the normalization factor. Mathematically, we model the cell count $n_{st}$ of an attribute-of-interest value $t$ and sample $s$, as a negative binomial (NB)

$$n_{st} \sim NB(\mu_{st}, \varphi_t) \tag{5}$$

where $\mu_{st}$ is the mean number and $\varphi_t$ is the dispersion parameter. We further assume that $\mu_{st} = \mu_{vrt} = f_{vt} n_{vr}$, where $f_{vt}$ is the fraction of cells of type $t$ when they have clustering attribute value $v$, and where $n_{vr}$ is the total number of cells derived from sample $s$ (which has clustering attribute value $v$ and $r$ is the replication index). We next assume that the log of $f_{vt}$ is a linear function of $v$, so that the final regression is of the form

$$\log \mu_{vrt} = \alpha_t + \beta_t \log[n_{vr}] + \gamma_t v_{vr} \tag{6}$$

$$\log \mu_{st} = \alpha_t + \beta_t \log[n_s] + \gamma_t v_s \tag{7}$$

Thus, given the cell counts $n_{st}$ we simply run a negative binomial regression (NBR) against the two covariates $v_s$ and $\log[n_s]$, to find out if the cell counts vary significantly with the clustering attribute value $v$.

The covariate $\log[n_s]$ plays the role of a normalization factor, accounting for the total number of cells contributed by sample $s$. Wald z-statistics and P-values of association are obtained from this NBR.

Benchmarking against Louvain is done by direct comparison of these statistics. Because the original Louvain algorithm is deterministic, whilst ELVAR is not (in ELVAR optimization is performed in a non-sequential random manner), for benchmarking we perform 100 distinct ELVAR runs, comparing the distribution of Wald-test statistics to the Louvain-derived one with a one-sided Wilcoxon rank sum test. Of note, further below we also describe how we benchmark ELVAR to a non-sequential randomized version of Louvain, where the Louvain output may also differ between runs.

### ELVAR pseudocode

Below is an outline of pseudocode for running ELVAR. We assume the scRNA-Seq data is encoded in a Seurat object *seu.o* with all the required meta-information, including a clustering cell attribute to be used in the clustering (clustering attribute "CA"), and a cell attribute of interest for DA-testing that we call "SA" (secondary attribute). The first step is to normalize the data and to build the k-nearest neighbor cell-cell graph:

Step-1 (Normalization and construction of cell-cell graph):
- *seu.o <- FindVariableFeatures(seu.o,selection.method="vst");*
- *seu.o <- ScaleData(seu.o,features=rownames(seu.o));*
- *seu.o <- RunPCA(seu.o);*
- *Elbowplot(seu.o); ### to determine number of significant PCs: topPC*

The choice of $k$ in specifying the number of nearest neighbors should be chosen sensibly. Typically the ratio of number of cells/k should be around 50, so assuming 1000 cells, $k$ should be around 20:
- *seu.o <- FindNeighbors(seu.o,dims=1:topPC,k.param=20);*
- *adj.m <- as.matrix(seu.o@graphs$RNA_nn); diag(adj.m) <- 0;*
- *gr.o <- graph.adjacency(adj.m,mode="undirected");*
- *vertexN.v <- names(V(gr.o));*
- *vertex_attr(gr.o,name="CA") <- seu.o@meta.data$CA;*
- *is.connected(gr.o); ### check graph is connected (if not, incrementally increase k).*

Step-2 (Estimate optimal purity parameter $a$):
- *aOPT <- SelOptAlpha(gr.o, nRuns=100);*

Step-3 (Inference of enriched communities with EVA):
- *eva.o <- Eva_partitions(gr.o,alpha=aOPT,Vattr.name="CA");*
- *comm.o <- ProcessEVA(eva.o,seu.o);*

Step-4 (Do DA-testing of attribute-of-interest with negative binomial regressions):
- *nbr.o <- DoDA(eva.o,seu.o,comm.o,DAattr="SA");*

The object *nbr.o* is typically the output of the *glm.nb* function from the *MASS* R-package, and statistics of association between the SA (e.g. Cd4t activation status) and CA (e.g. age) attributes can be extracted using *summary(nbr.o)$coeff*. Of note, to run ELVAR, the following R-packages need to be installed: *Seurat_4.1.1, SeuratObject_4.1.0, igraph_1.3.3, MASS_7.3-58.2, dplyr_1.0.9, Matrix_1.5-1*.

### Benchmarking ELVAR against Louvain

Because EVA is a direct extension of Louvain, it is natural to benchmark ELVAR against an analogous algorithm that uses the Louvain algorithm in place of EVA. The ordinary Louvain algorithm, as originally implemented by Blondel et al.[18] gives a deterministic network partition output, i.e. every run of the Louvain algorithm results in the same partition. This arises because during an optimization run, nodes are visited and assessed for local moving sequentially. This deterministic

"sequential" version of Louvain ("LVdet") was implemented in the versions of the *igraph* R package older than 1.3.3. On the other hand, EVA builds upon algorithmic details implemented in the Leiden algorithm[37], which results in potentially different partitions every time EVA is run. Specifically, similar to the Leiden algorithm, during an optimization run in EVA, we consider random (non-sequential) selection of nodes, which can therefore result in a different partition every time EVA is run. Thus, when benchmarking ELVAR against the Louvain-analog, it is important to account for these implementation differences. We do this by benchmarking against a non-sequential version of Louvain (as implemented in the current 1.3.3 version of igraph), where during an optimization run, nodes are visited randomly, which may also result in different partitions for different runs. In this work, we denote the algorithm analogous to ELVAR that uses the non-sequential Louvain in place of EVA, "LVnonseq".

### Simulation model benchmarking EVA against Louvain

First, we benchmarked EVA against Louvain with a simulation model in order to validate our novel R-implementation of EVA. We selected 200 classical monocyte cells from the TMS lung tissue scRNA-Seq 10X dataset, ensuring all cells derive from the same mouse (mouseID=19) and thus from the same age. For 100 of these cells, we then modified their scRNA-Seq profiles, simulating a "perturbed" cell-state, as follows. We randomly selected 50 genes among all genes not expressed in any of the 200 cells. For each perturbed cell, we then randomly subselected 20 genes from these 50, whose values were then altered in the cell, by randomly drawing 20 non-zero expression values from the distribution of non-zero expression values of the whole data matrix. Thus, this procedure generates a weak but significant co-expression structure among the 100 perturbed cells. Seurat was then applied to the 20138 gene ×200 cell scRNA-Seq data matrix, with VST feature selection followed by PCA. Top-8 PCs were selected to build the k-nearest neighbor cell-cell graph using k = 6. Louvain clustering algorithm as implemented in igraph was used to infer communities. EVA was run on the same cell-cell graph using a cell's perturbation state as the clustering attribute. Since the EVA result depends on initialization, we performed a total of 100 runs for each of nine choices of purity index parameter $a$ ($a$ = 0.1, 0.2, …, 0.8, 0.9). The final value of $a$ was chosen heuristically as the value at which purity increased compared to the Louvain solution ($a$ = 0) without compromising modularity too much. The quality of the EVA and Louvain clustering was assessed using the Adjusted Rand Index against the cell's perturbation state, as well as using Chi-Square statistics.

### Simulation model benchmarking ELVAR against Louvain-analog

In order to benchmark ELVAR against an analogous algorithm that uses Louvain in place of EVA, we generalized the previous simulation model. We selected all classical monocytes from mice with mouse-IDs 0 and 1 representing 1 month old mice (201 & 284 cells), mouse IDs 2 and 3 representing 3 month old mice (51 & 60 cells), mouse IDs 13 and 14 representing 21 month old mice (104 & 138 cells) and mouse IDs 21 and 22 representing 30 month old mice (94 & 61 cells). For young mice (1 and 3 month old mice), 25% of cells were perturbed using the same procedure described previously. For old mice (21 and 30 months), the frequency of perturbed cells was increased to 50%. Thus, this model simulates an age-related increased in a perturbed state. The cell-cell graph was derived as before, this time using the top 10 PCs and $k$ = 20, i.e $k$ was increased in line with the larger number of cells ($n$ = 993). ELVAR and the analogous Louvain-based algorithms were run on this cell-cell graph, in this case using age as the clustering attribute information. As before, EVA was initially run a 100 times for each of nine choices of $a$ parameter, in order to select an optimal $a$ based on overall purity and modularity values. Using the optimal $a$ value, ELVAR was then compared to the Louvain-analog (using both deterministic and non-sequential versions of Louvain) in its ability to predict the

increased frequency of the perturbed cell-state (here, the attribute of interest) in the older mice.

## Application of ELVAR to detecting shifts in lung tissue Cd4+ T-cell subtypes

As part of the 10X lung-tissue TMS set, a total of 551 Cd4+ T-cells were profiled to allow testing of a shift in naïve to mature subtypes with age. We removed cells from 4 mice each contributing less than 10 cells, leaving a total of 537 cells from 11 mice representing five age-groups: 143 (1 m), 122 (3 m), 67 (18 m), 107 (21 m) and 98 (30 m). Cells expressing Lef1, a well-known marker of naïve Cd4+ T-cells[22], were defined as naïve ($n = 186$), the rest as mature ($n = 351$). ELVAR was then applied to determine if the naïve/mature proportions change with age. Cell-cell graph was constructed using Seurat with VST and 8 top PCs and $k = 10$. EVA was run a total of 100 times with $a = 0.8$ (the optimal value in this dataset).

## Application of ELVAR to M1/M2 polarization analysis in lung alveolar macrophages

As part of the 10X lung-tissue TMS set, lung alveolar macrophages were abundantly profiled ($n = 1261$) to allow testing of a shift in M1/M2 macrophage polarization with age. We removed cells from mice displaying batch effects, leaving a total of 1124 cells from 15 mice representing five age-groups: 517 (1 m), 184 (3 m), 193 (18 m), 91 (21 m) and 139 (30 m). In order to annotate these 1124 lung alveolar macrophages into M1/M2 subtypes, we first identified 5 robust murine M1 (Cd80, Cd86, Fpr2, Tlr2, Cd40) and 5 robust M2 markers (Egr2, Myc, Arg1, Mrc1, Cd163) from the literature[38]. In an initial annotation, we declared cells as M1 if they co-expressed at least 2 of the 5 M1 markers, and similarly for M2. Cells annotated to both M1 and M2 subtypes were re-assigned an undetermined (UD) category alongside all other cells not annotated to either M1 and M2, resulting in 308 M1, 195 M2 and 621 UD-cells. We reasoned that UD-cells clustering predominantly with either M1 or M2 cells could be re-assigned to M1/M2 subtypes. To this end, we developed an iterative algorithm that reassigns the status of UD-cells to either M1 and M2, depending on their relative proportions among the neighbors of a given UD-cell. In more detail, we used the cell-cell graph as inferred using Seurat, and a multinomial test with $P < 0.05$ threshold to identify the UD-cells whose polarization status could be reassigned to either M1 or M2 status. We also required the absolute difference between the proportion of M1 and M2 neighbors of a given UD-cell to be larger than 0.2. This procedure was iterated 20 times, but numbers already converged after 7 iterations, resulting in 464 M1, 214 M2 and 446 UD-cells. ELVAR was then applied to determine if the M1/M2 proportions change with age. Cell-cell graph was constructed using Seurat with VST and 9 top PCs and $k = 20$. EVA was run a total of 100 times with $a = 0.7$ (the optimal value in this dataset).

## Application of ELVAR to detect DA-shifts in the nasopharynx of COVID-19 patients

We analyzed the scRNA-Seq dataset of Chua et al.[27], which profiled nasopharyngeal swabs from moderate and critically ill COVID-19 patients. We applied ELVAR to investigate if fractions of immune cells change from moderate to critical COVID-19 cases. We downloaded the Seurat data object from FigShare (https://doi.org/10.6084/m9.figshare.12436517) which contains QC-processed data in addition to the count matrix and metadata tables associated with cell type, patient identification and disease severity. The data matrix encompassed 80,109 immune cells derived from 8 moderate and 11 critical COVID-19 patients. In our analysis, we discarded one moderate sample (BIH-Cov-18) having only 23 immune cells and the immune cell-states MC, MoD-Ma, NK, and pDC each containing only fewer than 1000 cells. We randomly picked 1500 cells per cell-type for each of the remaining 9 cell-types (B cell, CTL - Cytotoxic T cell, moDC – monocyte-derived dendritic cell, Neu – Neutrophil, NKT - NKT cell, NKT-p - Proliferating

NKT cell, nrMa - Non-resident macrophage, rMa - Resident macrophage, Treg - Regulatory T cell), resulting in a total of 13,500 cells, of which 9005 cells derived from the 7 moderate patients and 4495 cells from the 11 critical patients. ELVAR was applied to each of the 9 cell-types to see if their abundance changes between moderate and severe Covid-19 cases. The cell-cell graph used as input to ELVAR was constructed using Seurat with variance stabilization for feature selection, and selecting the top 30 PCs with $k = 50$. ELVAR was run a total of 100 times with $a = 0.8$ and COVID-19 severity (moderate and critical) in the patients as the clustering cell attribute for community detection.

## Application of ELVAR to detect DA associated with smell loss post COVID-19 infection

We applied ELVAR to investigate changes in cellular composition within the olfactory epithelium that are associated with post-acute sequelae of COVID-19 infection (PASC), specifically by comparison of normosmic controls to hyposmic patients (long-term smell loss). The scRNA-Seq count matrices were downloaded from GEO under accession numbers GSE201620, GSE184117 and GSE139522[28,39,40]. In our analysis, we included 5 control samples with smell identification values (SIT) > 26, characterizing them as normosmic, and 5 hyposmic PASC patients. Seurat was used to normalize and batch-correct the data with the following variables (percentage of mitochondrial gens, patient condition and patient ID), following the procedure of Finlay et al.[28], and to perform cluster analysis, defining 35 cell clusters. To identify cluster specific marker genes, we applied the Seurat function FindAll-Markers(only.pos = TRUE, min.pct = 0.25, logfc.threshold = 0.5) and annotated the clusters associate with the olfactory epithelium (sustentacular cells, olfactory sensory neurons cells, olfactory horizontal basal cells (HBCs), Bowman's gland cells, microvillar cells) using the cell-state specific marker genes from Durante et al.[40]. From the Seurat clusters, we extracted only olfactory epithelium cells (11,173 cells), encompassing 8918 normal control cells and 2255 PASC cells for downstream ELVAR analysis. The distribution of cell-states was 1863 sustentacular-cells, 1428 sensory neurons-cells, 1901 HBCs-cells, 4561 Bowman's gland-cells and 1420 microvillar-cells. A cell-cell graph was constructed using Seurat with variance stabilization for feature selection, and selecting the top 40 PCs with $k = 40$. ELVAR was run a total of 100 times with $a = 0.7$ and using the smell loss phenotype (normal control and PASC) as the clustering cell-attribute for community detection.

## Application of ELVAR to colon cancer progression

We applied ELVAR to explore if the fractions of epithelial stem-cells and T-regulatory cells changes with disease progression. The analysis was performed in two ways. In the first approach, we downloaded Seurat objects from https://github.com/winstonbecker/scCRC_continuum which contain QC-processed data and cell-type annotations for a subset of samples. In the case of epithelial cells, the analysis was performed on a subset of the data consisting of stem-cells, TA2 & TA1 transit amplifying progenitors, enterocyte progenitors, immature enterocytes and differentiated enterocytes cell states. We randomly picked 1000 cells from each cell state (thus a total of 6000 cells) in order to reduce the computational runtime because two cell states contained ≥30k cells. Next, we removed cells from 2 donors each contributing less than 10 cells and cells from 1 donor displaying a batch effect, leaving a total of 5810 cells from 11 donors representing four disease stages: 1153 (Normal), 1672 (Unaffected), 2911 (Polyp) and 74 (Adenocarcinoma). The distribution of cell-states was 843 stem-cells, 971 TA2, 1000 TA1, 999 enterocyte progenitors, 998 immature enterocytes and 999 enterocytes number of cells. A cell-cell graph was constructed using Seurat with variance stabilization for feature selection, and selecting the 15 top PCs with $k = 20$. ELVAR was run a total of 100 times with $a = 0.8$ and disease stage as the main cell-attribute for community detection. For the analysis of T-regulatory (Tregs) cells we

focused on the subset of data consisting of Tregs, NK, Naïve T, CD4+ and CD8+ cells. We removed cells from 3 donors each contributing less than 10 cells and cells from 1 donor displaying a batch effect, resulting in a total of 6171 cells from 8 donors representing four disease stage groups: 381 (Normal), 3721 (Unaffected), 1900 (Polyp) and 169 (Adenocarcinoma). The distribution of cells across cell-types was: 472 Tregs, 245 NK, 1042 Naïve T, 3063 CD4+ and 1349 CD8+ number of cells. The cell-cell similarity graph was constructed using Seurat with variance stabilization for feature selection, selecting the 15 top PCs and number of nearest neighbors $k = 20$. ELVAR was run a total of 100 times with $a = 0.8$ with disease stage as the clustering attribute.

In the second approach, we downloaded the raw count snRNA-Seq matrices from GEO (GSE201348). Data was normalized using Seurat with variance stabilization for feature selection, leaving a total of 380,527 cells. Because the cell-type annotation for the full dataset was not provided, we applied dimensional reduction, clustering, UMAP visualization and well-known marker genes from Becker et al.[29] to the cells from the normal samples only, to annotate well separated cell clusters into enterocyte, goblet, immune-cell, stromal and endothelial cell categories. We then used Wilcoxon tests and marker-specificity scores[30,31] to build an mRNA expression reference matrix for these 5 broad cell categories. With this mRNA expression reference matrix in place, we then used our robust partial correlation framework[31] to estimate cell-type probabilities for all cells from all disease stages. Using a probability threshold of >0.7, we were thus able to confidently annotate 1866 endothelial cells, 104,009 enterocyte cells, 78,421 goblet cells, 24,973 immune-cells and 7941 stromal cells. Because of the very high sparsity of the snRNA-Seq data, in order to confidently identify stem-like cells among the 104,009 enterocytes, we applied our validated CancerStemID algorithm[32,33] which approximates stemness of single-cells from the estimated differentiation activities of tissue-specific transcription factors. In this instance, we used a set of 56 colon-specific TFs and their associated regulons, already validated by us previously[33]. The regulons were applied to the snRNA-Seq data, to estimate transcription factor differentiation activity (TFA) for each of the 56 TFs in each of the 104,009 enterocyte cells. We then declared stem-cell like cells as those displaying average TFA levels over the 56 TFs less than a threshold given by the 5% quantile of the average TFA distribution defined over the normal cells only. For ELVAR analysis, we only retained samples contributing at least 50 cells. For all other samples, all cells up to a maximum of 500 randomly selected cells were chosen, resulting in a total of 31,385 cells, drawn from 69 samples (8 normal, 16 unaffected FAPs, 41 polyps and 4 CRCs), encompassing 3761 normal, 7443 unaffected, 18,558 polyp and 1623 CRC cells. The cell-cell $k = 50$ nearest neighbor graph was constructed using Seurat, and ELVAR run a 100 times with $a = 0.8$, using disease stage as the clustering attribute.

## Comparison of ELVAR to DA-seq and Milo

We wish to compare the three algorithms in their ability to detect differential abundance of an attribute of interest relative to the clustering attribute. In our context the clustering attribute is the biological condition such as age or disease-status. The attribute of interest will refer to e.g. Cd4t activation status, cell-type or differentiation stage. We note that Milo[10] and DA-seq[7] are primarily designed to detect differential enrichment of one particular cell attribute in regions of the single-cell manifold, and hence need to be extended to allow for DA-testing of one cell attribute relative to another. What the three algorithms have in common is the inference of groups of cells that display differential enrichment relative to one particular cell attribute (the clustering attribute or biological condition). The methods differ in how these groups of cells are inferred. In ELVAR, we use the clustering attribute information when inferring cellular communities from the nearest neighbor cell-cell graph, subsequently identifying those that display enrichment for any specific clustering attribute value

(e.g. age-group). In contrast, DA-seq and Milo infer local regions, or potentially overlapping cellular neighborhoods, displaying significant enrichment of the biological condition (e.g. age-group). Thus, one way to compare all three algorithms for downstream DA-testing of an attribute of interest relative to the biological condition, is by first selecting the cells that appear in these significant communities/ regions/neighborhoods, and subsequently running negative binomial regressions of cell counts vs biological condition, taking biological replicates into account and normalizing for the total number of cells that each replicate sample contributes, as described earlier for ELVAR. In effect, once you have selected the cells within an enriched cluster (ELVAR) or an enriched local neighborhood (DA-seq, Milo), the subsequent strategy of running NBRs is unchanged and exactly the same for all three methods.

To understand the difference in performance between methods, we developed the following metric. Methods may display different sensitivity to detect DA of an attribute of interest relative to the clustering attribute because the significantly associated cell groups derived from each method (i.e. age-group enriched communities in the case of ELVAR, age-associated neighborhoods/regions in the case of Milo/DA-seq) may capture different numbers of cells. To make this clear, consider a scenario where one of the methods (call it "X") can't detect a cell group with sufficient numbers of old cells, say it detects a cell-group with at most 10 old cells, with 6 of these belonging to one cell-state "A", with the remaining 4 belonging to another cell-state "B". In contrast, another method "Y" does infer a large enough cell-group consisting of old-cells, say 30 cells with 15 belong to state "A" and 15 belonging to state "B". As far as young cells are concerned, all methods are able to infer a cell-group with a considerable number of cells, say 50 young cells, with 40 belonging to state "A" and 10 belonging to state "B". Because method "X" was not able to identify a cell-group with sufficient numbers of old-cells, it lacks power to detect the relative decrease of state "A" with age (two-tailed Fisher-test $P = 0.22$), whilst method "Y" has the power to detect it (two-tailed Fisher-test, $P = 0.006$). Whilst this hypothetical example ignores the variation due to sample replicates or variations due to replicate cell numbers, it clearly illustrates that the fraction of captured cells (fCaptCells) per clustering attribute value will strongly influence a method's power to detect DA of an underlying cell-state (the attribute of interest) with respect to this clustering cell attribute. Mathematically, we define the fraction of captured cells per clustering attribute value $a$ and from method $m$ by:

$$fCaptCells_{ma} = \frac{|(\#Cells\ with\ attribute\ value = a) \cap (\#Cells\ in\ groups\ from\ method\ m)|}{(\#Cells\ with\ attribute\ value = a)}$$

(8)

Specifically, a method that attains higher $fCaptCells_{ma}$ across the whole range of clustering attribute values $a$, including the extremes if the attribute is ordinal, will display higher power.

## Evaluation of ELVAR's robustness to batch effects and false positives

Robustness to batch effects and false positives in cell-type annotation was assessed in the four main datasets and in relation to the following cell-types that displayed significant DA: (i) in the TMS lung-tissue dataset we considered the robustness of the decreased naïve and and increased mature Cd4t cell fractions with age, (ii) in the Chua et al. Covid-19 nasopharynx dataset we considered the robustness of the increased neutrophil and decreased moDC cell fractions with Covid-19 disease severity (mild vs severe), (iii) in the Finlay et al. olfactory epithelium Covid-19 dataset we considered the robustness of the increased microvillar and decreased sensory-neuron cell-fractions with long-term smell loss, (iv) in the colon polyp and cancer snRNA-Seq dataset, we considered the increased enterocyte stem-cell and

decreased different enterocyte fractions with disease progression. In the case of batch effects, we compared ELVAR's performance in each of the four datasets in three different scenarios: (a) no batch/sample correction, (b) batch/sample correction with Harmony[34] and (c) batch/sample correction with Seurat[19]. We note that in all cases batch refers to the sample (i.e. mouse or individual). In the case of Harmony and Seurat, we thus used the sample-ID to perform the batch correction over, inferring in each case a new and different cell-cell graph. In each scenario we ran ELVAR a total of 100 times recording for each run the Wald z-statistics from the NBRs and corresponding cell-types. We also recorded in each case the likelihood ratio test P-values and sign of the regression coefficient, from which an alternative z-statistic was then derived using normal quantiles. For cell-types and studies where the Wald z-statistic breaks down (e.g. when for a given cell-type there are zero counts for all samples of a given biological condition) we used the likelihood ratio test P-values and derived z-statistics.

To assess robustness under an increased false positive rate (FPR) in cell-type annotation, we simulated false positives in each study by randomly re-annotating cells of the given cell type of interest with the label of another cell-type. Since the clustering step in ELVAR does not depend on the cell-type annotation, robustness was assessed at the NBR-step, by performing 100 Monte-Carlo re-annotations and subsequently computing the sensitivity to detect the DA-shift of the cell-type of interest over these 100 runs. This sensitivity was computed for increased values of the FPR (0.05, 0.1, 0.2, 0.3, 0.4 and 0.5).

To assess robustness under false positives in the clustering attribute (i.e. in the sample phenotype annotation), we simulated a small realistic ~20% fraction of misdiagnosed cases. We restricted this analysis to the two Covid-19 and colon-polyp datasets because errors in disease diagnosis are possible, and because the number of samples in each phenotype was sufficiently large to adequately model an ~20% misdiagnosis rate. Specifically, in the Covid-19 disease severity dataset, we randomly flipped 2 severe with 2 moderate cases, i.e. a total of 4 FPs among the 18 samples. In the Covid-19 smell-loss dataset, we randomly flipped 1 case and control, amounting to a total of 2 FPs among the 10 samples. In the former case, we performed 50 distinct randomizations, whilst in the latter there were at most only 25 (5 cases and 5 controls) distinct combinations of FPs. For the colon-polyp set, we randomly permuted 3 sample labels, performing 50 distinct permutation. For each label randomization/permutation, we ran ELVAR a total of 100 times, recording in each run the negative binomial regression Wald z-statistics of association between cell-type fractions and the phenotype. We estimated sensitivity as the fraction of runs where a significant change ($P < 0.05$) in the cell-type fraction was detected (preserving the same directionality of change as in the unperturbed scenario).

### Reporting summary
Further information on research design is available in the Nature Portfolio Reporting Summary linked to this article.

## Data availability
The snRNA-Seq dataset of colon cancer progression is publicly available from GEO (www.ncbi.nlm.nih.gov/geo/) under accession number GSE201348. The TMS scRNA-Seq data is available from https://doi.org/10.6084/m9.figshare.8273102.v2. The scRNA-Seq dataset from Chua et al. was downloaded from figshare (https://doi.org/10.6084/m9.figshare.12436517). The scRNA-Seq data from Finlay et al. is available from GEO under accession numbers GSE201620, GSE184117 and GSE139522. Source data are provided with this paper.

## Code availability
ELVAR is freely available as an R-package from https://github.com/aet21/ELVAR and published on figshare https://doi.org/10.6084/m9.figshare.22787498.

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

## Acknowledgements
This work was supported by NSFC (National Science Foundation of China) grants, grant numbers 31970632 and 32170652. We would like to thank the TMS consortium for making their data open-access and to everyone who supports open-access data.

## Author contributions
Study was conceived by A.E.T. Statistical and computational analyses were performed by A.K.M. and A.E.T. Manuscript was written by A.E.T. with contributions from A.K.M.

## Competing interests
The authors declare no competing interests.
