## [Peer Review File · Nature Communications]

Cell-attribute aware community detection improves differential abundance testing from single-cell RNA-Seq dataReviewers' comments:

Reviewer #1 (Remarks to the Author):

Overall, the authors have proposed yet another method for scRNA-seq data clustering. From the methodology perspective, the method components are pretty standard. From the result perspective, beyond performance verifications and comparisons with other methods, case studies have been conducted. However, novel insights into both methodology and molecular genetics are lacking. Therefore, I conceive that the manuscript should be submitted to a more specialized journal such as Bioinformatics or PLOS Computational Biology, but not Nature Communications. My concerns are as follows:

1. One can easily add extra information or data into the existing clustering process for improvement. Therefore, the addition of cell attributes is not surprising. For such a task in fine-dividing cell types further, existing hierarchical clustering methods (there are many, single-linkage, complete-linkage, average-linkage with different cell/gene similarity metrics.....) can already be sufficient in fine-dividing cell types beyond major cell types. It all just depends on how we cut the resultant hierarchy by an arbitrary threshold for high-resolution cell type clustering or low-resolution cell type clustering.
2. For the results, the proposed method is lack of sufficient comparisons (at the Nature Communications level) with other existing scRNA-seq data clustering methods.
3. The robustness across different sequencing data platforms are also speculative.
4. The software link should be clearly put at the end of abstract, similar to the existing bioinformatics paper.
5. There are many parameter settings along the proposed method. The existing parameter analysis experiments are not sufficient.

Reviewer #2 (Remarks to the Author):

The authors developed a differential abundance (DA) testing pipeline, ELVAR, which incorporates cell attribute information during community detection for scRNA-Seq analysis. They first validated the proposed pipeline using simulated scRNA-Seq data, where they showed improved clustering results to the Louvain algorithm with a mild increase in runtime. ELVAR was further applied to Cd4+ T-cells in lung and alveolar macrophages to investigate age-associated DA, and ELVAR successfully captured the subtle age-related DA patterns by observing decreased Cd4+ T-cell and M1/M2 ratio with age. Other DA testing methods (DAseq and MiloR) were compared, and ELVAR showed higher sensitivity to other methods. ELVAR was then applied to human snRNA-Seq data, and solely detected the increased stem-cell and T-cell fractions with cancer development despite the sparsity of the data. There are several issues need to be addressed or clarified before publication.

Major comments:

1. I wonder whether the authors can propose a systematic criterion determining the purity index parameter (α) rather than a subjective decision that can vary case by case. Also, most of the experiments used a fairly large α (0.7-0.9), and I am curious whether the clustering can be biased too much toward purity due to this large α . Please add a discussion on possible problems when purity is too much weighted compared to the modularity with a large α value. For example, it should be checked whether the clustering is not ignoring the mRNA profile too much.

2. The results are promising in terms of sensitivity, which casts a doubt on how ELVAR handles the robustness against false positive cell state information. False positives are expected when defining cell state (e.g., cancer stem cells) based on expression levels of markers. How robust the pipeline is to different rates of false positives in cell state by comparing the results with increasing the false positives in cell state. For example, different numbers of randomly chosen cancer stem cells can be added to the dataset and the DA results can be then compared.

3. Some results arouse doubt on whether data normalization and batch correction are done thoroughly (e.g. Figure 2a). This should be clarified in Results because normalization and batch correction removes systematically inter-sample variations, which are also handled by ELVAR. I wonder whether the results differ with and without batch correction and whether batch correction increases the sensitivity of ELVAR.

Minor comments:

1. Figure 2c (right panel): Is an adjusted P-value threshold used for the binomial test?

2. Figure 3b: Please add the objective function Z against the purity index parameter α . Also, provide a more thorough rationale for selecting $\alpha=0.8$ where both purity and modularity drop. Assuming the Z pattern over α based on P and Q patterns, it seems to me that $\alpha=0.7$ is better than 0.8. According to the descriptions in Results, the authors chose $\alpha=0.8$ based on the ratio of cluster numbers produced by EVA to those by classical Louvain. What is the rationale behind this?

3. Figure 4, 5e, 5j: When comparing ELVAR to MiloR, the enriched neighborhood is used. Please add a discussion on what would be the reason for the failure of MiloR although it also uses cells from the neighborhoods enriched with conditions (i.e., some of MiloR runs resulted in the opposite z values- e.g., z(Naive~Age) in Fig 4b).

4. Figure 2c (right panel): add 'n.s.' (see Figure 3d)

5. Figure 2d, add N/P ratio.

6. Figure 3a (right panel): 'naive' to 'naïve'

7. Page 4, Line 126: add an acronym for Tabula Muris Senis(TMS), as it is abbreviated afterward.

Reviewer #3 (Remarks to the Author):

The authors present an alternative approach to differential abundance testing in scRNA-seq data that uses sub-clustering to identify higher-resolution cell states followed by statistical testing. For the clustering step the authors propose using EVA, an extension of the commonly used Louvain graph-based clustering algorithm that considers node attributes as well as the graph structure. The testing step consists of a negative binomial regression on the cell counts associated with the variable of interest. The method is provided in the ELVAR R package and represents a midpoint between existing clustering-based DA methods (which usually act on broader cell types) and unsupervised methods which consider the local neighbourhood structure.

The method is demonstrated on a range of real and simulated datasets including the lung

immune system and colorectal cancer where it identifies expected shifts in cell states associated with covariates such as age and disease status. While I find the approach interesting I have some comments and concerns.

Concerns and comments

- I found it unclear what the main claim of the paper is meant to be, either that EVA provides more specific/useful clusters or that doing DA on subclusters is informative. Even in the abstract you state that ELVAR is a "novel DA-testing paradigm" but then compare it to Louvain which is a clustering algorithm. It would be very helpful to the reader if this was clarified and these two points delineated.
- I think you have somewhat oversold the novelty of the approach. This is not an entirely new algorithm, instead it is two commonly used processes (clustering and regression) applied at a higher resolution. This is still useful but the text implies this is an entirely new method which is not the case.
- I was concerned about the reliance on the "a" parameter but you do a good of showing how that can be chosen and I was relieved to see a function for choosing this in the package.
- You compare to Daseq and MiloR which are unsupervised DA methods. While this is interesting the more direct comparison would be to your approach but using Louvain rather than EVA. This would reveal whether it is the sub-clustering approach that is important or the choice of clustering method. I would also be interested to see how using some of the single-cell DA tests (scCODA, propeller) for the testing step compares to NBRs but that is less important.
- You mention the run time of the approach a few times but it is unclear if this is for a single run or normal usage (which requires many runs to tune the a parameter). Waiting 10 minutes to get results is not an issue but if it is 10 minutes per run that quickly becomes impractical. Please clarify this in the text. It would also be useful to note the runtimes of the other methods to put this in context.
- I found the references to "cell state" as an attribute that could be used for clustering confusing. To me, cell state is something that is defined after clustering, annotation etc. while what you use are things like age and disease status which are metadata of the sample known in advance. If you want to keep this terminology I suggest clarifying it in the text.
- You mention that you had to modify Daseq and MiloR for your use case but I couldn't find in the methods what those modifications were. Please add that information (or point out where it is if I missed it).
- I found the term "main attribute" a bit unclear and much prefer "clustering attribute" which is used in the methods so I would suggest using that throughout the text. It would also be clearer to use something like "attribute of interest" instead of "secondary attribute".
- You state that clustering beyond broad cell types is not possible with current commonly-used methods. While I agree this is non-trivial it is often done successfully so I don't feel this is a fair statement.
- There is some inconsistency in how other method names (Daseq, MiloR) are written/capitalised. Please make sure these are consistent and match how they are written by the method authors.

Responses to Reviewer's Points:

Reviewer #1 (Remarks to the Author):

General Comment: Overall, the authors have proposed yet another method for scRNA-seq data clustering. From the methodology perspective, the method components are pretty standard. From the result perspective, beyond performance verifications and comparisons with other methods, case studies have been conducted. However, novel insights into both methodology and molecular genetics are lacking. Therefore, I conceive that the manuscript should be submitted to a more specialized journal such as Bioinformatics or PLOS Computational Biology, but not Nature Communications.

Response: We would like to thank the reviewer for taking time to read and assess our MS. We opine that this MS deserves higher visibility than that afforded by a more specialized journal, for the following reasons. First of all, our manuscript is not about clustering, so it is wrong for the reviewer to imply that “all we have done is to present yet another clustering method for scRNA-Seq data”. Indeed, our MS presents a method for differential abundance (DA) testing, which is a completely different task to clustering. DA-testing tries to assess if the abundance of specific cell-types change in relation to a factor such as aging or disease. Clustering is about identifying and defining cell-types, so these are clearly two distinct tasks. Hence, whilst we acknowledge that our DA-testing method does include a clustering step, this clustering is not an endpoint of our algorithm, nor are we using this clustering step to define cell-types. In other words, we are using clustering in an entirely different and novel way to identify more subtle cell-states within cell-types, which has absolutely nothing to do with identifying clusters of cell-types as this reviewer has been incorrectly implying.

We would also like to remind this reviewer that not many DA-testing methods have been published. The paper by Dann and Marioni presenting Milo was only published in 2022 and they compared their algorithm to only a few other DA-testing methods, simply because DA-testing is an understudied problem in the scRNA-Seq field. Hence, our MS presenting ELVAR is both timely and fills an important gap in the literature: the cell-attribute aware clustering approach we have presented has not been explored before in the omic data analysis field, so how can the reviewer dismiss our work as not being methodological novel? In fact, we have implemented the cell-attribute aware clustering algorithm from scratch using R-software, and that has some significant merit.

Finally, the reviewer also claims that there is no novel biology in this MS. Whilst our manuscript is mainly methodological and thus not aimed at presenting novel biology, there are a number of important biological insights that our MS delivers. First, we demonstrate using snRNA-Seq data that polyps contain a higher fraction of stem-like and T-regulatory cells compared to normal colon

tissue. Previously, the attempt to infer this from snRNA-Seq data had failed and it had only been shown with scATAC-Seq data. Our successful application of ELVAR demonstrates that the insight is obtainable from snRNA-Seq data, without the need for scATAC-Seq. This is an important insight for many groups in the single-cell community who are thinking of generating single-cell omic data in precancerous lesions and who may be wondering what data-type might be best to profile. Indeed, the quantification of stem-cell fractions in precancerous lesions could be predictive of future cancer risk, so assessing which data-type might be best suited to quantify is a critically important question for the biomedical cancer field.

In this MS we also demonstrate that the naïve Cd4t cell fraction decreases with age in lung tissue. Whilst this confirms known results in blood, it is important to demonstrate this in the context of solid tissues like lung. We have now also analyzed two additional Covid-19 related scRNA-Seq datasets, demonstrating that e.g. ELVAR can detect a lower sensory neuron fraction in the olfactory epithelium of Covid-19 patients experiencing long-term smell-loss (shown in new SuppFigs.S5-S6). Whilst this is confirmatory, it is important to have validated this finding using a more rigorous approach.

Major points:

Comment 1. One can easily add extra information or data into the existing clustering process for improvement. Therefore, the addition of cell attributes is not surprising. For such a task in fine-dividing cell types further, existing hierarchical clustering methods (there are many, single-linkage, complete-linkage, average-linkage with different cell/gene similarity metrics.....) can already be sufficient in fine-dividing cell types beyond major cell types. It all just depends on how we cut the resultant hierarchy by an arbitrary threshold for high-resolution cell type clustering or low-resolution cell type clustering.

Response: We appreciate the reviewer's point but do not understand how this is relevant. Does the single-cell community use hierarchical clustering for scRNA-Seq data? The answer here is clearly no. The most popular clustering approaches use graph-based methods, as exemplified by the Louvain algorithm as implemented in the Seurat package. Hence, it is natural to consider improvements to Louvain, and the cell-attribute aware EVA algorithm that we consider is precisely a natural extension of Louvain, which is one reason why we use it. Instead of acknowledging that we have spotted an important novel way to do clustering within a graph-based framework, this reviewer dismisses our original and novel insight as "unsurprising", yet any alternative this reviewer offers (different flavours of extremely old hierarchical clustering) are much older methods that are not being used at all within the single-cell community.

Comment 2. For the results, the proposed method is lack of sufficient comparisons (at the Nature Communications level) with other existing scRNA-seq data clustering methods.

Response: We strongly disagree with this assessment. First, let us point out that there are not many published DA-testing methods. In our MS, we compare ELVAR to three other DA-methods, which includes two competing and recently published DA-testing methods (Milo and DA-Seq), of which Milo (published in 2022) could be considered the current state-of-the-art, and a third method analogous to ELVAR that uses Louvain instead of EVA, which has not been published and which alongside ELVAR is entirely novel. Milo itself was only compared to 2 other DA-testing algorithms, so we feel that a benchmarking of our ELVAR algorithm against the state-of-the-art (Milo), DA-Seq and an analogous method based on Louvain, is entirely adequate. Indeed, these comparisons allow us to rigorously benchmark ELVAR's performance in relation to methods that do not use clustering (Milo & DA-seq use fuzzier representations known as cellular neighborhoods) and a clustering-based method that does not use cell-attribute-aware community detection (the "Louvain-analogue" mentioned above), thus allowing us to assess the importance of the cell-attribute aware clustering step. Of note, we have dedicated a whole Fig.5 to a comparison of ELVAR to Milo and DA-Seq, and the comparison to the "Louvain-analogue" is displayed in Fig.2e, Fig.3g, Fig.4 and SuppFigs.S4-S6. That there are not more DA-testing algorithms to compare to simply reflects the fact that this is a relatively novel topic and that new DA-testing methods (like ours) need to be developed.

Comment 3. The robustness across different sequencing data platforms are also speculative.

Response: Again, this comment suggests that the reviewer did not read our MS carefully enough. In fact, we not only demonstrate that ELVAR works on 10X scRNA-Seq data, but also on single-nucleus RNA-Seq (snRNA-Seq) data. Importantly, the original study analysing samples in colon cancer progression failed to identify an increased stem-cell fraction in polyps from the snRNA-Seq data, and had to rely on scATAC-Seq to make this inference. ELVAR was able to clearly detect the increase of the stem-cell fraction in polyps from the snRNA-Seq itself. In our opinion, this addresses the reviewer's concern. Moreover, given that the simpler algorithms such as Louvain have been shown to work well irrespective of scRNA-Seq platform, it seems very unreasonable from this reviewer to suggest that extensions of Louvain would not exhibit the same robustness. In other words, we feel that this is relatively minor point given the substantial evidence that existing clustering or DA-testing algorithms work well irrespective of scRNA-Seq platform.

Comment 4. The software link should be clearly put at the end of abstract, similar to the existing bioinformatics paper.

Response: This is an editorial matter and we are happy to do so if the editor so wishes. The link to our software is here: <https://github.com/aet21/ELVAR> and this is clearly specified in the “Code availability” section as required by journals.

Comment 5. There are many parameter settings along the proposed method. The existing parameter analysis experiments are not sufficient.

Response: We appreciate the reviewer’s point, but in fact our algorithm is relatively parameter-free in comparison to other proposed methods, which is therefore one appealing feature of our method. Indeed, there is only one main parameter within the clustering step of ELVAR, which is the purity parameter “a” that controls the relative weighting of purity over modularity when maximising the objective function. As we demonstrate throughout the manuscript (Fig.2b, Fig.3b and SuppFlg.S1) this parameter also acts as a quasi-resolution parameter, controlling the size of the inferred modules. Hence, as far as this “a” parameter is concerned we do provide extensive analyses (Fig.2b, Fig.3b and SuppFlg.S1), addressing the key question of how “a” affects module size as well as the modularity and purity of the modules. We now also provide a clear quantitative criterion of how “a” should be chosen in practice, described in a new Methods section, and have correspondingly also added a new function “SelOptAlpha” to our R-package. In addition, in SuppFig.S2 we also make a detailed comparison of EVA to Louvain in relation to the traditional resolution parameter “gamma, γ ”. We note that this resolution parameter is set to 1 in the original Louvain algorithm but that it can also be varied to further modulate module sizes. In summary, we feel that we have adequately addressed this issue, as indeed acknowledged by Reviewer-3 who categorically stated that “I was concerned about the reliance on the “a” parameter but you do a good job of showing how that can be chosen and I was relieved to see a function for choosing this in the package.”

Reviewer #2 (Remarks to the Author):

General Comment: The authors developed a differential abundance (DA) testing pipeline, ELVAR, which incorporates cell attribute information during community detection for scRNA-Seq analysis. They first validated the proposed pipeline using simulated scRNA-Seq data, where they showed improved clustering results to the Louvain algorithm with a mild increase in runtime. ELVAR was further applied to Cd4+ T-cells in lung and alveolar macrophages to investigate age-associated DA, and ELVAR successfully captured the subtle age-related DA patterns by observing decreased

Cd4+ T-cell and M1/M2 ratio with age. Other DA testing methods (DAseq and MiloR) were compared, and ELVAR showed higher sensitivity to other methods. ELVAR was then applied to human snRNA-Seq data, and solely detected the increased stem-cell and T-cell fractions with cancer development despite the sparsity of the data. There are several issues need to be addressed or clarified before publication.

Response: We would like to thank the reviewer for taking time to read and evaluate our manuscript, and for the constructive comments, which have helped us to significantly improve the MS.

Major comments:

Comment 1. I wonder whether the authors can propose a systematic criterion determining the purity index parameter (a) rather than a subjective decision that can vary case by case. Also, most of the experiments used a fairly large a (0.7-0.9), and I am curious whether the clustering can be biased too much toward purity due to this large a . Please add a discussion on possible problems when purity is too much weighted compared to the modularity with a large a value. For example, it should be checked whether the clustering is not ignoring the mRNA profile too much.

Response: We thank the reviewer for raising this excellent point. In response to this, in the Methods section of the revised version of the manuscript we now provide clear quantitative criteria as to how the optimal “ a ” parameter value should be chosen. In addition, we now provide a novel R-function “SelOptAlpha” within our ELVAR R-package, dedicated to choosing the optimal “ a ” parameter value. We note that in the original EVA paper, the parameter “ a ” was chosen based on the relative number of inferred clusters as well as the maximum cluster size, all measured relative to the corresponding values obtained using the ordinary Louvain algorithm. We modified this criterion, because in our opinion the maximum cluster size can be highly variable for a non-deterministic optimization algorithm such as EVA. Our modification entails studying at which value of the “ a ” parameter, the number of inferred clusters is significantly greater than that predicted by the Louvain algorithm, whilst also requiring relatively high purity without much degradation to the modularity. The new quantitative criteria that we now provide in Methods as well as the SelOptAlpha function are a big improvement and we thank the reviewer for pointing this out to us.

As to the other concern expressed by the reviewer, let us clarify that the optimal a -value being in the range 0.7-0.9 (and therefore “close” to the value 1) does not mean that “purity overly influences the clustering solution”. In fact, it is important to understand that the purity index P is very much constrained by the cell-cell graph structure, with this cell-cell nearest neighbour graph being derived solely from the scRNA-Seq data. The reviewer seems to have been misled into thinking that the cell-cell nearest neighbour graph is influenced by the purity parameter, but this is not the case: the cell-cell nearest neighbour graph is derived first from the scRNA-Seq data. Hence, the

topology of this network is entirely determined by the cells' RNA profiles. We did not use any other cell attribute information when building this cell-cell network. However, once the graph is built, we do use other cell-attribute information to infer the communities/clusters within this graph. For this reason, the purity and modularity of the clustering solutions display remarkable stability under variations in the parameter "a" (as shown in e.g. Fig.2b), except when "a" enters the range 0.7-0.9. Typically, the optimal "a" parameter is chosen at the point where we start to observe deviations in the number of inferred clusters relative to the number obtained using the Louvain algorithm, hence this is a parameter regime that is actually quite "far" from the regime $a=1$ where modularity is ignored. In response to this reviewer's point, we have now clarified this important issue in Discussion.

Comment 2. The results are promising in terms of sensitivity, which casts a doubt on how ELVAR handles the robustness against false positive cell state information. False positives are expected when defining cell state (e.g., cancer stem cells) based on expression levels of markers. How robust the pipeline is to different rates of false positives in cell state by comparing the results with increasing the false positives in cell state. For example, different numbers of randomly chosen cancer stem cells can be added to the dataset and the DA results can be then compared.

Response: We thank the reviewer for raising another good point. First of all we need to clarify to the reviewer that in this manuscript we always run ELVAR with a clustering attribute that is known with certainty. For instance, in the case of aging, the cell attribute used in the clustering is the age of the mouse which is known without error. Likewise, in the colon dataset, the clustering attribute of the cell is the stage of disease (normal, polyp, adenoma), which is known. So when testing for DA, we are testing for a shift in the abundance of another cell-attribute (cell-type or cell-state) in relation to this clustering attribute. As such, false positives that may arise because of the uncertainty in calling cell-type/state would not affect the ELVAR clustering solution. In effect, the reviewer's concern only applies to the second step in ELVAR where we run negative binomial regressions to assess if cell-type/state frequency changes as the clustering attribute value is altered. For this reason, ELVAR is quite robust to false positives arising from the definition of cell-type, as long as the FPR is below a certain threshold, as demonstrated in the figure below for four of the scRNA-Seq datasets analyzed in the revised version of the MS:

Figure legend: Left panel: Sensitivity to detect a significant (NBR P -value < 0.05) change in the abundance of naïve and mature Cd4T-cells with age in the lung tissue of mice under different rates of false positives (FPR, x-axis). Sensitivity was estimated over 100 distinct runs. Middle left panel: As left panel, but for detecting a significant change in the abundance of neutrophils and monocyte-derived dendritic cells (moDC) with Covid-19 disease severity. Middle right panel: As other panels, but for detecting a significant change in the abundance of olfactory sensory neurons and microvillar cells with Covid-19 smell-loss phenotype. Right panel: as other panels, but for detecting a significant change in the abundance of enterocyte stem and differentiated cells with colorectal adenoma progression.

The above figure, which was generated following the reviewer's suggestion, has been incorporated into a new Figure-6e in the revised version of the MS.

As a side note, we should also clarify that in the case of the colon dataset, we performed the analysis in two ways to test robustness against cell-type annotation. In one case, we used the cell-type annotation provided by the authors which relies on marker expression. In the second case, we did not use expression markers to call (cancer) stem cells. Instead, we used our marker-free single-cell entropy method to identify the stem cells and cancer stem cells, which is actually more robust than single expression markers, see e.g. Teschendorff & Enver Nat Commun 2017, Chen W, Teschendorff A Commun Biol.2019). Nevertheless, even with our entropy method, we acknowledge that there will be a significant number of FPs, and the simulation analysis above (as suggested by the reviewer) shows that the final result is very robust to FPs as long as the FPR is not too high.

Comment 3. Some results arouse doubt on whether data normalization and batch correction are done thoroughly (e.g. Figure 2a). This should be clarified in Results because normalization and batch correction removes systematically inter-sample variations, which are also handled by ELVAR. I wonder whether the results differ with and without batch correction and whether batch correction increases the sensitivity of ELVAR.

Response: This is another excellent point and we apologize for not having mentioned or clarified this issue appropriately. Indeed, because ELVAR clusters cells using cell-attribute information into account (specifically cells belonging to the same age-group or disease state), this can help

circumvent sample-specific batch effects which would otherwise cause cells of the same attribute to segregate. We take the reviewer's point on board in that (i) we had not explicitly mentioned whether prior batch correction had been performed and (ii) we had not explored if batch correction affects ELVAR's performance. In the revised version of the manuscript, we have now made a comprehensive evaluation of ELVAR's performance in 4 separate scRNA-Seq datasets without batch correction and with prior batch correction using two different batch correction methods (Harmony- Korsunsky et al Nat Methods 2019 and Seurat- Butler et al Nat Biotech.2018). The result is displayed in the figure below, which we have incorporated into the revised version of the MS as a new main Figure 6a-d:

Figure legend: **a)** Violin plots of negative binomial regression z-statistics of association of cell-type counts with age for mature and naïve lung-tissue Cd4t cells and for 3 different scenarios: None=ELVAR was run with no batch (sample) correction, Harmony=ELVAR was run data batch-corrected with Harmony, Seurat: ELVAR was run on data batch-corrected with Seurat. Each violin plot contains the values for 100 distinct ELVAR runs. The P-values are derived from two-tailed Wilcoxon rank sum tests comparing “None” to either “Harmony” or “Seurat”. Horizontal dashed lines indicate the $P=0.05$ significance level. **b)** As a) but for nasopharyngeal neutrophils and monocyte-derived dendritic cells (moDC) cell counts changing with Covid-19 disease severity (mild vs critical). **c)** As a) but for microvillar and sensory olfactory neurons in the olfactory epithelium (OE) in relation to their counts changing with long-term smell loss in Covid-19 patients. **d)** As a) but for colon enterocyte stem-cell and differentiated cell counts changing with cancer stage progression.

This figure shows that in general we don't gain much by performing prior batch correction using either Harmony or Seurat. In fact, the figure demonstrates that results (i.e. significance vs non-significance) are relatively insensitive to whether we do prior batch correction or not, and that the degree of statistical significance is generally speaking stronger when we run ELVAR without prior batch correction, although results are study-specific. For instance, for OE:Sensory Neurons, the

significance is stronger after Seurat-batch-correction, but sensitivity is unaffected, probably because this was a case where the biological signal was strong. In general, we can understand the results of the figure as follows: any good normalization and batch-correction strategy would aim to increase the signal-to-noise ratio (SNR). If we perform batch correction across samples (e.g. patients, mice) it is inevitable that this procedure reduces biological signal (“S”) if the samples differ by the biological condition of interest (e.g. age, disease state/stage). The only reason why batch correction could improve the SNR in the DA-testing context, is that it also reduces the technical noise term (“N”) by a larger amount, resulting in an increased SNR. In the context of the datasets analyzed in this work it is clear to us that batch correction via Harmony or Seurat could remove more biological signal in scenarios where the biological signal is weak (e.g. see panel a) in above figure), resulting in a lower SNR, which may explain why the subsequent DA-testing leads to lower levels of statistical significance. Because ELVAR uses the biological condition (e.g. age or disease state/stage) as the clustering attribute, this can definitely help circumvent any batch effects that may exist, because the cell-attribute aware clustering can “draw in together” cells from different samples but the same biological condition. In addition to adding the above figure as a new main figure, we have added a new subsection to Results presenting and discussing these findings.

Let us also clarify here that for all the datasets analyzed in this work, we always used the QC-processed data as provided by the authors of the respective publications, as this facilitates comparability of results. Thus, if we don’t explicitly mention that batch-effect correction was performed, then this means that no additional correction was performed. For instance, in the case of Fig.2a, the data derives from the Tabula Muris Senis (TMS) (10X scRNA-Seq, Nature 2020), which is a dataset that we had already analyzed previously (see Maity AK et al Nat Aging 2022), and where we showed that batch effects are of a lesser magnitude than the changes associated with age. The same conclusion was reached by the Nature TMS paper (Nature 2020, 583, pages590–595) itself, which did not do any batch correction across mice, because doing so removes important biological variability associated with age. Indeed, as far as Fig.2a is concerned, we disagree that this displays a batch effect. For instance, in Fig.2a we can see that the cells from the two 30-month old mice are clustering somewhat separately from the rest. Had it just been one of the mice we may worry about a batch effect, but since the cells from the two older mice are clustering closer together, this suggests a biological effect. In any case, Fig.2a also demonstrates that there is an overall lack of clustering structure which is actually a good thing because if cells were forming clear separate clusters labelled by mouseID we would then infer the presence of big batch effects. The intermingling of cells from different mice and similar age-groups suggests that there is no major batch effect.

Minor comments:

Comment 1. Figure 2c (right panel): Is an adjusted P-value threshold used for the binomial test?

Response: We thank the reviewer for asking this. Yes, we always use Bonferroni corrected P-values (hence very conservative) throughout to call statistical significance. In other words, in the right panel of Fig.2c the colored entries in the confusion matrix denote significance under Bonferroni adjustment, whilst the color-bar denotes unadjusted P-values. We note that in Fig.1c, which is the figure depicting the general method, we do specify that we use Bonferroni, and this then applies to all subsequent figures including Fig.2c (right panel). In response to the reviewer's point, we have now clarified in the legend to Fig.2c that we use Bonferroni adjusted P-values for calling statistical significance.

Comment 2. Figure 3b: Please add the objective function Z against the purity index parameter a. Also, provide a more thorough rationale for selecting a=0.8 where both purity and modularity drop. Assuming the Z pattern over a based on P and Q patterns, it seems to me that a=0.7 is better than 0.8. According to the descriptions in Results, the authors chose a=0.8 based on the ratio of cluster numbers produced by EVA to those by classical Louvain. What is the rationale behind this?

Response: We thank the reviewer for raising this good point. Reason for not having included Z in Fig.3b is simply that in the original EVA paper Z is not used when selecting the optimal "a" parameter value, so we did not want to give the false impression that "Z" plays an important role in this parameter selection process. However, in response to the reviewer's point we have now included Z in Fig.3b.

The reason why $a=0.8$ in Fig.3b was chosen is as follows: First, we need to point out that the parameter "a" influences the number of inferred clusters and therefore also cluster-size, i.e. the more clusters we infer these are generally also of smaller size (higher resolution). Ideally (following the recommendations of the original EVA paper) we want to see a departure from the ordinary Louvain solution in terms of the number of clusters and cluster size. We found it very useful to plot the number of inferred clusters relative to the Louvain solution (i.e. the ratio), because this signals the parameter regime where we start to get "something different" compared to Louvain. Indeed, it is worth noting here that the authors of the original EVA paper provide a criterion for selecting "a" which is entirely based on the number of inferred clusters (relative to Louvain) and the maximum community size (also relative to Louvain). However, we did not like this metric, because the maximum community size is highly variable for stochastic optimization algorithms such as the one implemented in EVA. For this reason, we decided to modify their criterion to also take into consideration not only the number of inferred clusters (relative to Louvain) but also the actual purity

and modularity values, which are also dependent on “a”. Specifically, as far as purity and modularity is concerned, we want a parameter value that yields a relatively large purity without much degradation in modularity. Thus, together with the number of inferred clusters, the criterion becomes one where we seek an a-value that leads to a tangible increase in the number of inferred clusters, with a relatively high purity and without much degradation in the modularity. Based on Fig.3b, the optimal a-value is thus 0.8. It is not 0.7 because for a=0.7, we don’t get an increase in the number of inferred clusters, so this choice would also go against the recommendations of the original EVA paper.

We acknowledge that our criterion for selecting the optimal a-value as presented in our MS still appears subjective, since we need to quantify what is an increased number of clusters, what is a relatively higher purity and how much degradation in the modularity should be allowed. For this reason, in the revised version of the MS we now provide a very concrete quantitative criterion for selecting the optimal a-value. This criterion is described in Methods, and we now also provide a new R-function called “SelOptAlpha”, which we have incorporated into our ELVAR R-package, and which helps select the optimal a-value.

Finally, we should also note that the precise value of the a parameter is not absolutely critical, as long as it falls within a specific region, a region that is normally in the range 0.7-0.9 for the datasets analysed here. Importantly, all our results are fairly robust to small changes in the a parameter within this 0.7-0.9 range.

Comment 3. Figure 4, 5e, 5j: When comparing ELVAR to MiloR, the enriched neighborhood is used. Please add a discussion on what would be the reason for the failure of MiloR although it also uses cells from the neighborhoods enriched with conditions (i.e., some of MiloR runs resulted in the opposite z values- e.g., z(Naive~Age) in Fig 4b).

Response: The reviewer has asked an excellent key question. The data previously displayed in Fig.4b, 5e and 5j have now been re-arranged and combined into a new Fig.5, which for convenience we display below:

Figure legend: **a)** Violin plots displaying the z-statistics of associations between cell-counts (y-axis, mature Cd4t-cells and naïve Cd4t-cells) and age derived from negative binomial regressions for ELVAR, Milo and DAseq, as shown. Grey dashed line indicates the threshold of statistical significance $P=0.05$. P-values derive from a one-tailed Wilcoxon rank sum test comparing ELVAR derived z-statistics to those from DA-seq and Milo, respectively. **b)** Violin plots displaying the fraction of captured mature and naïve Cd4t cells (fCC[Naive+Mat], y-axis) for each biological condition (x-axis) and method. **c-d)** As a-b) but for alveolar macrophage M1 and M2 subtypes. **e-f)** As a-b), but for stem-cell and T-regulatory cell fractions in colon tissue in relation to colon cancer progression (N=normal, U=unaffected FAP cases, P=polyp, A=adenoma).

To address the reviewer's question and to understand why ELVAR is doing better, we need to inspect panels b,d,f in the above figure. What we can see is that ELVAR captures many more cells belonging to a given biological condition (age-group or disease stage) that are part of a community that is enriched for cells of that biological condition, as compared to a method like Milo. As to why Milo displays a lower sensitivity, this is because Milo is "unsupervised" i.e. it builds the cellular neighborhoods without regard to the biological condition (e.g. age-group). In contrast, ELVAR uses the biological condition as an attribute when inferring communities, and so this leads to a higher sensitivity. In the case of age, because the changes associated with age are of a small effect size, using the age-attribute information can circumvent the noise in the scRNA-Seq data that would otherwise inevitably obscure an underlying community structure. Another key observation from panel b above, is that the improved sensitivity of ELVAR over Milo is higher for the older age-group, which is biologically intuitive because older cells would display higher heterogeneity and are therefore less likely to share the same cellular neighborhood as inferred with Milo. With ELVAR,

because it uses age-attribute information into account, these older cells are more likely to be “brought together” within the same cellular community.

As to the opposite z-values which Milo displayed for a number of runs, the reviewer should note that these z-statistic values are not statistically significant. Whenever two-sided statistics are not statistically significant, we expect them to be distributed around zero with both positive and negative statistics.

We note that the above explanation as to why ELVAR displays a higher sensitivity than Milo was already mentioned in Discussion, but in response to the reviewer’s point we have also expanded this in the relevant paragraph of Discussion.

Comment 4. Figure 2c (right panel): add ‘n.s.’ (see Figure 3d)

Response: We thank the reviewer for pointing this out to us. We have now added “n.s” to Fig.2c.

Comment 5. Figure 2d, add N/P ratio.

Response: As requested, we have added this information to Fig.2d.

Comment 6. Figure 3a (right panel): ‘naive’ to ‘naïve

Response: Whilst Word may automatically modify the letter “i” to ï so as to reflect the French pronunciation, it is perfectly acceptable in English to write the word “naïve” also as “naive”. In the actual figure, we are using a software that does not seem to support this particular diacritic, so we are unable to do this modification at this stage, but will look into it if the editor so requires.

Comment 7. Page 4, Line 126: add an acronym for Tabula Muris Senis(TMS), as it is abbreviated afterward.

Response: We thank the author for pointing this out to us. We have now added the acronym.

Reviewer #3 (Remarks to the Author):

General Comment: The authors present an alternative approach to differential abundance testing in scRNA-seq data that uses sub-clustering to identify higher-resolution cell states followed by statistical testing. For the clustering step the authors propose using EVA, an extension of the commonly used Louvain graph-based clustering algorithm that considers node attributes as well as the graph structure. The testing step consists of a negative binomial regression on the cell counts associated with the variable of interest. The method is provided in the ELVAR R package and

represents a midpoint between existing clustering-based DA methods (which usually act on broader cell types) and unsupervised methods which consider the local neighbourhood structure.

Response: We would like to thank the reviewer for taking time to read and evaluate our manuscript and for the constructive comments, which have helped us improve the manuscript. We are delighted that this reviewer has grasped the essence of ELVAR, as indeed it is a midpoint between clustering-based and local neighbourhood DA-methods. We felt that exploring this midpoint would be important and that cell-attribute aware clustering would be an interesting approach to consider.

General Comment: The method is demonstrated on a range of real and simulated datasets including the lung immune system and colorectal cancer where it identifies expected shifts in cell states associated with covariates such as age and disease status. While I find the approach interesting I have some comments and concerns.

Response: We are delighted that the reviewer agrees that this is an interesting approach to consider.

Concerns and comments:

Comment-1: I found it unclear what the main claim of the paper is meant to be, either that EVA provides more specific/useful clusters or that doing DA on subclusters is informative. Even in the abstract you state that ELVAR is a "novel DA-testing paradigm" but then compare it to Louvain which is a clustering algorithm. It would be very helpful to the reader if this was clarified and these two points delineated.

Response: We thank the reviewer for raising this important point and apologize if the main claim was unclear. To clarify, our main claim is that using cell-attribute information when clustering cells can reveal communities that improve power for subsequent DA-testing, i.e. communities inferred using cell-attribute-aware clustering improve the sensitivity to detect DA-shifts compared to (i) an analogous method that does not use cell-attribute-aware clustering (we often refer to this analogous method as "Louvain-analogue", "Louvain-benchmark" or simply "LVnonseq" because it uses the non-sequential Louvain algorithm in place of EVA) and (ii) to methods based on inferring local neighbourhoods (DA-seq & Milo).

Our manuscript deals mainly with scenarios where the DA is being tested in relation to factors that only induce relatively small changes, e.g. for instance age-related shifts in CD4T-cell substates (naïve vs mature), or cancer progression-associated shifts in differentiation states within colon-tissue. That we provide an extensive benchmark of ELVAR to an analogous method that uses Louvain-clustering in place of EVA is critically important, because our hypothesis is that using the cell-attribute information (which is what EVA does and what Louvain does NOT do) will lead to

more relevant communities for subsequent DA-testing. Hence it is therefore very natural and sensible to benchmark our DA-testing method to one which uses Louvain instead of EVA. However, in this manuscript we obviously also need to compare our DA-testing method to competing DA-testing methods like Milo and DA-Seq which are not clustering-based. So, in effect, by comparing ELVAR to (i) an analogous method that uses Louvain instead of EVA and (ii) to non-clustering approaches such as Milo and DA-Seq, we address the two key questions: (a) Does cell-attribute information improve community-detection for subsequent DA-testing compared to a clustering based method that does not use cell-attribute information?, and (b) Does cell-attribute aware clustering lead to improved DA-testing compared to a non-clustering based approach? We think that this is rigorous and sensible, and we are only sorry that in our desire to be rigorous we have rendered the presentation unclear in so far as to what the main claim of the manuscript is. We suspect that part of the confusion could be due to the fact that we have interchangeably used the names EVA and ELVAR. Let us clarify here that although the acronym of our DA-testing algorithm (ELVAR) only reflects the clustering part, ELVAR refers to the whole algorithm encompassing both the clustering as well as the subsequent negative binomial regressions for DA-testing. As such we should have used “ELVAR” in Fig.2e and Fig.3g and we apologize for having incorrectly labelled that panel with “EVA”. In response to the reviewer’s point, we have now rewritten part of the Abstract, Introduction, Results and Discussion sections to further clarify the important points above, and have also corrected the labels in Fig.2e and Fig.3g.

Comment-2: I think you have somewhat oversold the novelty of the approach. This is not an entirely new algorithm, instead it is two commonly used processes (clustering and regression) applied at a higher resolution. This is still useful but the text implies this is an entirely new method which is not the case.

Response: We appreciate the reviewer’s point. Whilst we agree that our DA-testing method consists of two broad steps (clustering and subsequent negative binomial regressions), the clustering step is entirely novel in the sense that it implements cell-attribute aware clustering, which to our knowledge is a concept that has not yet been explored in the context of scRNA-Seq data, and certainly not in the context of DA-testing. In our opinion, this is therefore a novel algorithm for DA-testing in scRNA-Seq data, because it explores a “midpoint” between traditional clustering-based DA-testing methods and local cellular neighbourhood methods like Milo and DA-Seq, an algorithmic space that has not been explored before (as indeed acknowledged by this reviewer). We feel that it is important to explore this “midpoint” and that our manuscript has merit in (i) coming up with the idea of using cell-attribute-aware clustering to explore this algorithmic space, (ii) in successfully implementing the EVA algorithm using R-software (this R-implementation was

previously not available to the community) and (iii) in demonstrating that cell-attribute-aware clustering can lead to improvements in sensitivity for DA-testing.

In response to the reviewer's point, we have now removed the adjective "novel" from the Abstract and Introduction. We only use it when referring to the R-implementation of EVA which is indeed a novel implementation.

Comment-3: I was concerned about the reliance on the "a" parameter but you do a good of showing how that can be chosen and I was relieved to see a function for choosing this in the package.

Response: We thank the reviewer for recognising the importance of the "a" parameter, as indeed this parameter not only controls the relative importance of purity over modularity in the inference process but indirectly also controls the resolution (i.e. number and size of inferred clusters) of the communities. In network science, it is well-known that the resolution parameter and any other parameters that influence the resolution are of critical importance, which is why we have dedicated several figure panels to justify the criteria for selecting optimal a parameter values. Given the importance of this parameter, in the Methods section of the revised version we now provide a concrete quantitative criterion for selecting the optimal a parameter value. In addition, we have a new R-function called "SelOptAlpha" for selecting this optimal a-value, which is now part of our updated ELVAR R-package.

Comment-4: You compare to Daseq and MiloR which are unsupervised DA methods. While this is interesting the more direct comparison would be to your approach but using Louvain rather than EVA. This would reveal whether it is the sub-clustering approach that is important or the choice of clustering method. I would also be interested to see how using some of the single-cell DA tests (scCODA, propeller) for the testing step compares to NBRs but that is less important.

Response: The reviewer has raised an excellent point, and this is exactly why we have dedicated a considerable number of figures to a direct comparison of our DA-testing method to the analogous DA-method obtained by replacing EVA with Louvain, precisely as suggested by this reviewer. Indeed, the previous old Fig.2e, Fig.3g, Figs.5d and 5i as well as SuppFigs.S1,S2,S4-S6 all compared ELVAR (which uses EVA) to an analogous DA-method where we use Louvain in place of EVA. This was done precisely in order to demonstrate that cell-attribute aware clustering leads to an improvement over the more standard clustering approach. Hence, we apologize to the reviewer if the meaning of "Louvain" in the previous figures was unclear. To address the reviewer's point we now clarify that the comparison of ELVAR to the analogous DA-testing algorithm that uses Louvain in place of EVA is shown in new Fig.2e, Fig.3g, Fig.4f and SuppFigs.S4-S6 , where it appears under the abbreviation "LVnonseq" (to indicate the use of the non-sequential version of

Louvain). In the revised version we have now clarified what this algorithm stands for in the respective figure legends.

The reviewer also mentions that it would be interesting to consider comparisons of scCODA and propeller to NBRs. We thank the reviewer for bringing our attention to these important works, which we now also describe and cite in the Introduction section of our MS. As the reviewer also points out though, these comparisons would be less important because these methods do not really work at the level of the single-cell manifold. Once cells have been annotated to cell types, methods like scCODA and propeller do not take the single-cell manifold into consideration, which as argued by others (e.g. Milo and DA-seq authors, and we also subscribe to this point of view) is not an optimal approach as it ignores potential heterogeneity within the single-cell populations. As such, the appropriate comparison of our ELVAR method are to Milo, DA-seq and the analogous DA-method based on Louvain instead of EVA, since all of these methods take the single-cell manifold structure into consideration. Hence, as hinted by the reviewer the only sensible comparison would be between propeller, scCODA and the negative binomial regression (NBR) component of our algorithm. However, this specific comparison was already made in the propeller manuscript, hence we don't see the value of repeating that analysis here.

Comment-5: You mention the run time of the approach a few times but it is unclear if this is for a single run or normal usage (which requires many runs to tune the a parameter). Waiting 10 minutes to get results is not an issue but if it is 10 minutes per run that quickly becomes impractical. Please clarify this in the text. It would also be useful to note the runtimes of the other methods to put this in context.

Response: We thank the reviewer for asking this. We have now clarified in Results and Discussion section that 1 ELVAR run for a typical 30k cells per cell-type would take around 15-20 minutes to complete. As the reviewer correctly points out, if we want to perform runs to find the optimal "a" parameter value, this would require 900 ELVAR runs in total (9 parameter values x 100 runs per parameter value). Whilst this may seem impractical, most laboratories these days have access to a HPC facility, i.e. access to a 100-node server is now very common. Under these circumstances, the estimated completion time for the whole task would be in the range 90-135 minutes, which in our opinion is very feasible.

As far as a comparison to the 3 other methods is concerned (i.e. the analogous algorithm with Louvain in place of EVA, DA-seq and Milo), we have now expanded SuppFig.S3 to display runtimes for all 4 methods, which for convenience we display below:

SI fig.S3: Runtime comparison of EVA. Runtimes for one run in minutes (y-axis) vs. the number of cells (#Cells, x-axis), as estimated using the enterocyte snRNA-Seq dataset encompassing a maximum of 103913 cells, and for 4 different methods as shown. In the case of EVA and Louvain, the number of cells defines the size of the cell-cell k -nn graph, where $k=50$ was kept fixed throughout. In the case of DA-seq, because it requires a binary phenotype, we ran 3 separate comparisons for DA (normal+unaffected vs polyp, polyp vs adenoma, and normal+unaffected vs adenoma). Of note, for EVA and Louvain the timing measures the time to find communities given the input graph. To make the comparison objective and fair, the timing for DAseq was taken as the time to run the functions `getDAcells` and `getDAregion`, which does the inference of enriched regions, analogous to finding the enriched communities in the case of EVA. Likewise, for Milo the timing was measured from the construction of neighborhoods (`makeNhoods` function), ending with the neighborhood annotation (`annotateNhoods` function) which identifies enriched neighborhoods. For detailed parameter choices, see Methods section. All runtimes were obtained on a Dell Precision Workstation with an Intel Xeon(R) CPU E3-1575M v5 @3GHz and 64 GB RAM. Although workstation comes with 8 processing cores, runtimes were obtained without parallelization.

As we can see from the above figure, EVA is faster than DA-seq (this is because the current implementation of DA-Seq only allows binary phenotypes, so multiple pairwise comparisons are needed for categorical phenotypes with more than 2 categories), but slower than Milo. EVA, DA-seq and Milo are all much slower compared to an algorithm analogous to ELVAR that uses Louvain instead of EVA (denoted “Louvain” in the above figure). This is not unexpected since Louvain is fully unsupervised. In summary, whilst there are clear speed advantages to using Louvain or Milo, these do display lower sensitivity, and hence given that ELVAR runtimes remain very feasible with standard workstations/HPC facilities, we feel that there is a good case for applying ELVAR in practice. We now make this point in Discussion.

Comment-6: I found the references to "cell state" as an attribute that could be used for clustering confusing. To me, cell state is something that is defined after clustering, annotation etc. while what you use are things like age and disease status which are metadata of the sample known in advance. If you want to keep this terminology I suggest clarifying it in the text.

Response: The reviewer is absolutely right and we sincerely apologize for any confusion which we may have caused. In this work we use factors such as age or disease status, which are known with certainty, as the main cell-attribute in the clustering (the clustering attribute). In the revised version we have now made it clear throughout the whole manuscript that the clustering attribute is a biological condition such as age or disease stage, and not cell-state.

Comment-7: You mention that you had to modify DAseq and MiloR for your use case but I couldn't find in the methods what those modifications were. Please add that information (or point out where it is if I missed it).

Response: We thank the reviewer for asking this and apologize for the confusion, which may have been caused by inappropriate terminology. To clarify, Milo and DA-seq are primarily designed to detect differential abundance in relation to one particular cell attribute, but need to be extended (this is what we had meant by "modified") if you wish to assess differential abundance of one cell-attribute relative to another. The extended versions of DAseq and Milo are described in the Methods subsection titled "Comparison of ELVAR to DA-seq and Milo", and in response to the reviewer's point, we have now clarified the relevant paragraph in that subsection, which for convenience we repeat here: We wish to compare the three algorithms in their ability to detect differential abundance of an attribute of interest relative to the clustering attribute. In our context the clustering attribute is the biological condition such as age or disease-status. The attribute of interest will refer to e.g. Cd4t activation status, cell-type or differentiation stage. We note that Milo and DA-seq are primarily designed to detect differential enrichment of one particular cell attribute in regions of the single-cell manifold, and hence need to be extended to allow for DA-testing of one cell attribute relative to another. What the three algorithms have in common is the inference of groups of cells that display differential enrichment relative to one particular cell attribute (the clustering attribute or biological condition). The methods differ in how these groups of cells are inferred. In ELVAR, we use the clustering attribute information when inferring cellular communities from the nearest neighbor cell-cell graph, subsequently identifying those that display enrichment for any specific clustering attribute value (e.g. age-group). In contrast, DA-seq and Milo infer local regions, or potentially overlapping cellular neighborhoods, displaying significant enrichment of the biological condition (e.g. age-group). Thus, one way to compare all three algorithms for downstream DA-testing of an attribute of interest relative to the biological condition, is by first

selecting the cells that appear in these significant communities/regions/neighborhoods, and subsequently running negative binomial regressions of cell counts vs biological condition, taking biological replicates into account and normalizing for the total number of cells that each replicate sample contributes, as described earlier for ELVAR. In effect, once you have selected the cells within an enriched cluster (ELVAR) or an enriched local neighborhood (DA-seq, Milo), the subsequent strategy of running NBRs is unchanged and exactly the same for all three methods.

Comment-8: I found the term "main attribute" a bit unclear and much prefer "clustering attribute" which is used in the methods so I would suggest using that throughout the text. It would also be clearer to use something like "attribute of interest" instead of "secondary attribute".

Response: We take the reviewer's point on board and in the revised version we use the terminology "clustering attribute" for the attribute used in EVA, and "attribute of interest" for the attribute that we want to test DA for.

Comment-9: You state that clustering beyond broad cell types is not possible with current commonly-used methods. While I agree this is non-trivial it is often done successfully so I don't feel this is a fair statement.

Response: We appreciate the reviewer's point but also feel that there has been a misunderstanding here. We agree that clustering methods for scRNA-Seq data have led to the identification of cell subtypes or rarer cell-types that go beyond the definition of broader cell-types. However, most often, the defining characteristic of these rarer cell-types is their lower proportion within the tissue of interest, and not necessarily that their differential expression changes are of a smaller magnitude. When we study the effect of say age on a cell population, the age distribution does not map well to clusters because the differential expression changes associated with aging are quite small. So, in this context, the fraction of older cells could be quite high but the effect sizes associated with aging are so small that the age distribution of cells does not map neatly to distinct clusters, as shown in our manuscript. Hence, when formulating our statements we were implicitly thinking of cell-states induced by factors with small effect sizes, as opposed to infrequent cellular states characterized by larger effect sizes. Thus, we hope that this clarifies the intended meaning, but in response to the reviewer's point we have also rephrased some of the relevant sentences in the Introduction.

Comment-10: There is some inconsistency in how other method names (DAseq, MiloR) are written/capitalised. Please make sure these are consistent and match how they are written by the method authors.

Response: In the revised version, we now use the original method names “DA-seq” and “Milo” throughout the manuscript. MiloR of course was the name of the R-package associated with Milo, and we apologize for having used that instead of “Milo”.

REVIEWER COMMENTS

Reviewer #1 (Remarks to the Author):

I would like to thank the authors for the clarifications.

However, I am still not convinced as the authors mentioned differential abundance (DA) testing had already been proposed and addressed in DA-seq 2021 [7] and Milo 2022 [10]. On the other hand, from the historical perspective of bioinformatics development, such a concept in differential testing has already been well-studied in the context of other domains such as differential gene expression analysis and differential gene ontology enrichment. One can easily derive a similar version in the context of scRNA-seq.

Since it is 2023 now, I am not quite convinced that the submission represents a significant but not incremental contribution to be published on Nature Communications. I still believe it is just another paper to be published on Bioinformatics / PLOS CB / BIB.

In addition, the following observations are found:

1. The authors claimed "In fact, we have implemented the cell attribute aware clustering algorithm from scratch using R-software, and that has some significant merit." Unfortunately, I found it counter-intuitive as most of the algorithms in the proposed approach are classic data science / machine learning components. Although the authors have programmed by themselves, the concepts are already there. From the scientific point of view, programming from scratch does not lead to any novelty in methodology.

2. Indeed, the proposed approach is not entirely novel. It is heavily built on EVA [34]. The authors have even spend a whole page (p.13-p.14) on describing EVA, a method published on "Applied Network Science" (Impact Factor TBD, JCI QUARTILE Q2-Q3): "Here we describe the recently published algorithm (EVA: Extended LouVain Algorithm) to identify homogeneous communities in a network with node attributes [34]" "

On p.14, The ELVAR algorithm. "Building on EVA, we developed an algorithm and R-package called ELVAR for differential abundance (DA) testing in scRNA-Seq data....."

Reviewer #2 (Remarks to the Author):

Minor comments:

The authors have addressed most of the comments raised by this reviewer. However, one concern remains unclear. In response to my inquiry about the robustness of FP in cell state, the authors clarified that the cluster attribute is known with certainty, and as such, there is no need to demonstrate its robustness to FP. However, one of cell attributes used is the stage of disease. In certain diseases, the stage of disease or histological subtypes could be unclear or inconsistent even when several pathologists examined the specimen, which may result in false positives for this cell attribute and thus affect the purity index that are used for clustering. In this regard, I think that it would be better to understand how robust clustering is against varying numbers of FPs. At least, this point (possible presence of false positive cell attributes and how robust the clustering would be against the false positive attributes) should be discussed.

Reviewer #3 (Remarks to the Author):

Thank you to the authors for their replies to my comments, I am happy with the changes you

have made and feel they adequately address my concerns. The focus and terminology of the paper are now clearer and the comparisons I had misunderstood in the original version are now better explained.

I will not that an extension of Milo for testing between conditions has recently been released as a preprint <https://www.biorxiv.org/content/10.1101/2023.03.08.531744v1>. I don't expect you to change to using this as it was not available when the work for this paper was done but it may be worth noting in the methods where you explain how testing between conditions was done for Milo/DA-seq.

Detailed response to reviewer comments:

Reviewer #2 (Remarks to the Author):

Minor comment: The authors have addressed most of the comments raised by this reviewer. However, one concern remains unclear. In response to my inquiry about the robustness of FP in cell state, the authors clarified that the cluster attribute is known with certainty, and as such, there is no need to demonstrate its robustness to FP. However, one of cell attributes used is the stage of disease. In certain diseases, the stage of disease or histological subtypes could be unclear or inconsistent even when several pathologists examined the specimen, which may result in false positives for this cell attribute and thus affect the purity index that are used for clustering. In this regard, I think that it would be better to understand how robust clustering is against varying numbers of FPs. At least, this point (possible presence of false positive cell attributes and how robust the clustering would be against the false positive attributes) should be discussed.

Response: We once again would like to thank the reviewer for taking time to re-evaluate our manuscript and for the positive encouraging feedback. Whilst the understanding is that the published datasets we have analyzed have already undergone extensive QC, we share the reviewer's concern that there could be FPs in the clinical annotation, for instance due to e.g. misdiagnosis, and that ensuing errors in the clustering attribute could affect ELVAR's performance. Of course, it should be pointed out that any diagnostic errors would affect any downstream analysis, including for instance a simple differential expression analysis between diagnostic groups, so clearly any downstream analysis task will be influenced by such errors. In order to address the reviewer's point, we have now performed an additional analysis introducing FPs in the clustering attribute and checking the performance metrics of ELVAR. We have now done this on 3 of the real scRNA-Seq datasets (the colon-polyp as well as the two Covid-19 sets) where errors in the clustering attribute could arise. Such misdiagnosis errors are typically not that frequent and generally lower than 20%, yet because of sample size restrictions, we have allowed the fraction of FPs to be as high as ~20%. The result has been incorporated into a new SuppFig.S8, which for convenience we display below:

SI fig.S8: Robustness of ELVAR to errors in clustering attribute. *a)* Violin plots of the Negative Binomial Regression Wald z-statistics of association between cell-type counts and Covid-19 disease state (y-axis) obtained after introducing false positives (FP) in the clustering attribute (disease state). For the two Covid-19 datasets (nasopharynx & olfactory epithelium-OE) the fraction of false positives was 22% and 20%, respectively. In the case of OE, we had 5 controls and 5 cases, so we generated 25 perturbations in which 1 control and 1 case were flipped to generate 20% FPs. For each such perturbation we performed a 100 ELVAR runs, so a total of 2500 runs. Similarly, for nasopharynx, we randomly flipped 2 cases and controls (corresponding to 22% FPs) a total of 50 times, each time performing 100 ELVAR runs, so a total of 5000 runs. For each tissue-type, we display the two main cell-types for which a DA-association had been found without FPs. The sign after the cell-type abbreviation indicates the directionality of the association observed without FPs. Grey dashed line corresponds to the line ± 1.96 ($P=0.05$). *b)* Barplot comparing the sensitivities of detecting a significant association (Wald z-statistic with $P<0.05$) with the same directionality as in the unperturbed case, as computed over the total number of runs. For comparison, we also display the sensitivities when no FPs are present. *c-d)* As *a-b)*, but for the colon-polyp dataset focusing on enterocyte stem-cell and T-regulatory cells.

As the reviewer can see, an approximate 20% FPs in the clustering attribute can lead to variable drops in sensitivity, depending on dataset and cell-type. For instance, for detecting an increased stem-cell fraction in enterocytes during colon cancer progression, the sensitivity only dropped by ~5%, but to detect the decrease of sensory neurons in the OE of Covid-19 patients experiencing long-term smell-loss, the sensitivity dropped by over 30%. Bearing in mind that the true number of FPs in the clustering attribute is likely to be less than 20%, and that the sensitivity at 20% FPs is still over 60% for all cell-types, this suggests that the DA-shifts identified with ELVAR are relatively

robust. In response to the reviewer's point, in the revised MS we have not only included the new SuppFig.S8 above, but also describe these results in Discussion.

Reviewer #3 (Remarks to the Author):

Comment: Thank you to the authors for their replies to my comments, I am happy with the changes you have made and feel they adequately address my concerns. The focus and terminology of the paper are now clearer and the comparisons I had misunderstood in the original version are now better explained.

Response: We would like to thank the reviewer again for taking time to evaluate our manuscript and for the very positive feedback. We are very happy that the revisions have clarified the issues raised and thank the reviewer for giving us the opportunity to improve the MS.

Comment: I will note that an extension of Milo for testing between conditions has recently been released as a preprint <https://www.biorxiv.org/content/10.1101/2023.03.08.531744v1>. I don't expect you to change to using this as it was not available when the work for this paper was done but it may be worth noting in the methods where you explain how testing between conditions was done for Milo/DA-seq.

Response: We thank the reviewer for bringing our attention to this very interesting preprint. In response to this, we now briefly mention this extension of Milo, called MiloDE, in Discussion, as indeed it would be interesting in future work to compare ELVAR to MiloDE in relation to downstream DE-tasks. For convenience, in Discussion we now write: "Indeed, we stress that due to the cell-attribute-aware clustering, ELVAR was able to capture more cells of a given biological condition within a community of cells enriched for that biological condition, compared to neighborhood approaches like Milo/DA-seq, thus increasing power to detect subtle DA-shifts. **It would be interesting to explore if this improved sensitivity would also be seen relative to miloDE [34], a recently proposed extension of Milo, which uses a 2nd order k-nearest neighbor graph approach to generate a state manifold that displays more homogenous neighborhoods. Indeed, miloDE has been shown to significantly increase power in downstream differential expression tasks.**"

Reviewer #1 (Remarks to the Author):

Comment: I would like to thank the authors for the clarifications. However, I am still not convinced as the authors mentioned differential abundance (DA) testing had already been proposed and addressed in DA-seq 2021 [7] and Milo 2022 [10]. On the other hand, from the historical

perspective of bioinformatics development, such a concept in differential testing has already been well-studied in the context of other domains such as differential gene expression analysis and differential gene ontology enrichment. One can easily derive a similar version in the context of scRNA-seq. Since it is 2023 now, I am not quite convinced that the submission represents a significant but not incremental contribution to be published on Nature Communications. I still believe it is just another paper to be published on Bioinformatics / PLOS CB / BIB.

Response: We would like to thank the reviewer for taking time to re-evaluate our MS. The DA-Seq and Milo methods were published in 2021 and 2022 respectively, and hence these are very recent methods. There is a clear need to develop additional improved methods that explore different dimensions in algorithmic space, and ELVAR does exactly that. Moreover, we also disagree with this reviewer's statement that methods for DA-testing in scRNA-Seq data are "like methods for differential expression testing". For instance, noise levels in scRNA-Seq are by definition much higher than in bulk-tissue, and this means that more robust inferences would require consideration of the single-cell manifold, which ordinary differential expression methods would ignore. Milo and DA-seq are concrete examples which clearly refute the reviewer's point.

Comment: In addition, the following observations are found:

The authors claimed "In fact, we have implemented the cell attribute aware clustering algorithm from scratch using R-software, and that has some significant merit." Unfortunately, I found it counter-intuitive as most of the algorithms in the proposed approach are classic data science / machine learning components. Although the authors have programmed by themselves, the concepts are already there. From the scientific point of view, programming from scratch does not lead to any novelty in methodology. Indeed, the proposed approach is not entirely novel. It is heavily built on EVA [34]. The authors have even spend a whole page (p.13-p.14) on describing EVA, a method published on "Applied Network Science" (Impact Factor TBD, JCI QUARTILE Q2-Q3): "Here we describe the recently published algorithm (EVA: Extended LouVain Algorithm) to identify homogeneous communities in a network with node attributes [34] "

Response: Although we understand where the reviewer is coming from, we feel that this is a very harsh comment, as such criticism would apply to hundreds if not thousands of papers published in the computational biology domain. Because cell-attribute aware clustering has NOT yet been explored in the context of DA-testing in scRNA data, the "logical step" is to first consider adapting existing cell-attribute aware clustering algorithms to this DA-testing problem. If we had found that existing methods do not lead to an improvement, we would then have been forced to abandon this approach or to develop an improved novel cell-attribute aware clustering algorithm. However, given that we did find an improvement by adapting EVA, it seems unnecessary at this point to develop an

improved novel cell-attribute aware clustering algorithm. We feel that the reviewer should understand that there is significant merit in having spotted the opportunity to apply cell-attribute aware clustering to DA-testing in scRNA-Seq data. We are convinced that our paper being published in Nat Commun will lead to the wider community embracing cell-attribute aware clustering as a tool to address many other challenges in computational biology.